**EMBO** *reports*

# PHF2-mediated H3K9me balance orchestrates heterochromatin stability and neural progenitor proliferation

Samuel Aguirre [1,6], Stella Pappa[1,6], Núria Serna-Pujol [1], Natalia Padilla [2], Simona Iacobucci [1], A Silvina Nacht[3,4], Guillermo P Vicent [1], Albert Jordan[1], Xavier de la Cruz[2,5] & Marian A Martínez-Balbás [1✉]

## Abstract

**Heterochromatin stability is crucial for progenitor proliferation during early neurogenesis. It relays on the maintenance of local hubs of H3K9me. However, understanding the formation of efficient localized levels of H3K9me remains limited. To address this question, we used neural stem cells to analyze the function of the H3K9me2 demethylase PHF2, which is crucial for progenitor proliferation. Through mass-spectroscopy and genome-wide assays, we show that PHF2 interacts with heterochromatin components and is enriched at pericentromeric heterochromatin (PcH) boundaries where it maintains transcriptional activity. This binding is essential for silencing the satellite repeats, preventing DNA damage and genome instability. PHF2's depletion increases the transcription of heterochromatic repeats, accompanied by a decrease in H3K9me3 levels and alterations in PcH organization. We further show that PHF2's PHD and catalytic domains are crucial for maintaining PcH stability, thereby safeguarding genome integrity. These results highlight the multifaceted nature of PHF2's functions in maintaining heterochromatin stability and regulating gene expression during neural development. Our study unravels the intricate relationship between heterochromatin stability and progenitor proliferation during mammalian neurogenesis.**

**Keywords** PHF2; Heterochromatin Integrity; Histone Demethylation; Neural Stem Cells; DNA Damage
**Subject Categories** Chromatin, Transcription & Genomics; Neuroscience

## Introduction

During development epigenetic regulators are essential to control transcriptional programs and to maintain heterochromatin integrity. Nevertheless, it remains unclear how these processes are interrelated and how they evolve in coordination throughout development. In vertebrate early neurogenesis, the histone demethylase PHF2 is indispensable for the proliferation of neuronal progenitors. There, PHF2 controls transcription of genes associated with cell cycle progression by keeping low levels of H3K9 methylation at promoters (Pappa et al, 2019). Belonging to the α-ketoglutarate-Fe(2)(+)-dependent dioxygenases class, PHF2 features a plant homeodomain (PHD), a Jumonji domain, binding specifically to K4-trimethylated histone 3 (H3K4me3) (Fortschegger and Shiekhattar, 2011; Wen et al, 2010) and an intrinsically disordered domain (IDR) (Vicioso-Mantis et al, 2022). Despite being reported to exhibit demethylase activity on monomethylated H3K9 in vivo, the recombinant protein displays no activity in vitro. PHF2 demonstrates the ability to demethylate both histones and non-histone proteins after phosphorylation by PKA and in complex with ARID5B (Baba et al, 2011). This complex mediates the demethylation of methylated ARID5B, directing the PHF2-ARID5B complex to promoters. At these promoters, PHF2 further mediates the demethylation of H3K9me2, leading to the activation of target gene transcription (Baba et al, 2011). PHF2 has been also shown to regulate promoters of genes involved in various metabolic disease pathways (Bricambert et al, 2018). Consistent with its role at promoters, PHF2 has been implicated in the regulation of cell cycle genes in chicken neural tube and in mouse neural stem cells (NSCs) to allow neural progenitor expansion at early neurogenesis (Pappa et al, 2019). Overall, multiple lines of research suggest that PHF2, in coordination with other proteins, facilitates H3K9me2 demethylation at specific gene promoters, contributing to the maintenance of an active state in different developmental contexts.

[1]Department of Structural and Molecular Biology, Instituto de Biología Molecular de Barcelona (IBMB), Consejo Superior de Investigaciones Científicas (CSIC), Barcelona 08028, Spain. [2]Vall d'Hebron Institute of Research (VHIR), Passeig de la Vall d'Hebron, 119, E-08035 Barcelona, Spain. [3]Center for Genomic Regulation (CRG), Barcelona Institute for Science and Technology (BIST), Barcelona, Spain. [4]Universitat Pompeu Fabra (UPF), Barcelona, Spain. [5]Institut Català per la Recerca i Estudis Avançats (ICREA), Barcelona 08018, Spain. [6]These authors contributed equally: Samuel Aguirre, Stella Pappa. ✉E-mail: mmbbmc@ibmb.csic.es

Interestingly, PHF2 depletion leads to DNA damage (Alonso-de Vega et al, 2020; Pappa et al, 2019), R-loops accumulation (Pappa et al, 2019), and genome instability suggesting that in addition to promoter regulation, PHF2 might be involved in other cellular regulatory processes. Although preserving genome stability is crucial for the proper proliferation and functioning of neural progenitors, the molecular causes underlying such instabilities remain elusive.

Our current study reveals that, in contrast to the established role of PHF2 in maintaining an active state at gene promoters, the loss of PHF2 leads to increased chromatin accessibility in certain genome regions. Specifically, heterochromatic pericentromeric satellite repeats show an increased transcription upon PHF2 depletion. Heterochromatin constitutes a substantial part of metazoan genomes (Ho et al, 2014; Roadmap Epigenomics et al, 2015) and plays a crucial role in silencing repetitive elements and preventing their recombination (Janssen et al, 2018; Padeken et al, 2015). The repression of heterochromatin involves specific histone post-translational modifications (Allshire and Madhani, 2018; Janssen et al, 2018), with satellite repeats marked by di- and tri-methylation on histone H3 at lysine 9 (H3K9me2/3) (Grewal, 2023; Martens et al, 2005). This histone mark is deposited by suppressor of variegation 3-9 homolog 1/2 (SUV39H1/H2) and recognized by heterochromatin protein 1 (HP1), which facilitates silencing through a read-write mechanism mediated by its chromodomain (Grewal, 2023; Martens et al, 2005). Once the threshold of H3K9me3 levels is reached, a feedforward cascade of Suv39H1/2 read-write activity propagates heterochromatin until the boundaries. The borders of heterochromatin function through various mechanisms, including recruiting histone-modifying activities, protecting euchromatic modifications, creating nucleosome-free regions, modifying chromatin dynamics through transcription, and establishing chromatin loops by linking DNA to nuclear structures (Grewal, 2023). Nevertheless, the molecular components and dynamics of heterochromatin boundaries, particularly in higher eukaryotes, remain poorly understood.

This study demonstrates that PHF2 plays a crucial role in silencing pericentric repeats. Based on the presented data, we propose that PHF2-mediated H3K9 demethylation throughout the genome and particularly at the boundaries where PHF2 is enriched, balances H3K9me3 levels ensuring the silencing of satellite repeats. In this manner, PHF2 emerges as a regulator preventing DNA damage accumulation and maintaining genome stability, essential for the expansion of progenitor cells. These findings contribute valuable insights into the regulatory mechanisms governing heterochromatic territories in neural progenitor cells.

## Results

### PHF2 interacts with heterochromatin components

The expansion of neural progenitor cells is a fundamental process occurring during early neurogenesis, enabling the generation of the complete repertoire of neural cells necessary for proper development. The histone demethylase PHF2 is essential to NSCs expansion in vitro and in vivo (Pappa et al, 2019). To elucidate the mechanistic underpinnings of PHF2's impact on progenitor proliferation, we conducted an investigation to identify its interacting partners in neural progenitors. For this purpose, we employed a well-established self-renewal model mouse NSCs (Fig. 1A) where the role of PHF2 in proliferation has been described. These NSCs were derived from the cortex of E12.5 mouse embryos. They proliferate and can generate wide range of differentiated neural cell types in culture (Currle et al, 2007; Estaras et al, 2012; Pappa et al, 2019; Pollard et al, 2006). Co-immunoprecipitation (CoIP) experiments targeting endogenous PHF2 were performed using IgG or PHF2 antibodies in control NSCs (shCT) and NSCs with PHF2 knockdown (KD) (shPHF2). To achieve PHF2 depletion, we employed lentivirus expressing shRNA specific to PHF2, resulting in a partial decrease in PHF2 levels (Fig. 1A) while leaving the expression of other family members, PHF8 and KIAA1718, unaffected (Appendix Fig. S1A). Proteins associated with the immunopellet were identified through mass spectrometry analysis (Appendix Table S1). Similarly, to what has been described by others (Bricambert et al, 2018; Kim et al, 2019) the data do not suggest interaction with any complete pathways but do reveal specific targets related to translation and RNA processing (RL14, RL29, RS8), transcription (SSRP1, BTF3), three-dimensional chromatin structure (Lamin B and SMC2), DNA repair (YBX1), and heterochromatin (HP1BP3). We specifically confirmed by coimmunoprecipitation (CoIP) experiments the interaction between PHF2 and heterochromatin protein 1 binding protein 3 (HP1BP3), a protein that interacts with HP1α and belongs to the H1 family (Garfinkel et al, 2015) (Fig. 1B). Interestingly, it has been previously reported the interaction between PHF2 and SUV39H1 (Shi et al, 2014); interaction which was also confirmed by CoIP (Appendix Fig. S1B). Both SUV39H1 and HP1BP3 proteins play a role in the establishment and maintenance of constitutive heterochromatin, particularly pericentromeric heterochromatin (PcH), which is involved in various cellular processes, including cell division and DNA replication. PcH organizes into chromo-centers, defined as prominent DNA-dense nuclear foci that contain clustered pericentromeric satellite DNA repeats from multiple chromosomes in interphase mouse cells (Brandle et al, 2022; Jones, 1970; Pardue and Gall, 1970). It is worth to mention that PHF2 has been previously implicated as a centromeric associated factor (Ohta et al, 2010).

Following the identification of interactions between PHF2 and heterochromatin components, we aimed to investigate the potential enrichment of PHF2 at heterochromatic genomic regions. To address this, we performed PHF2 chromatin immunoprecipitation followed by sequencing (ChIP-seq) experiments in triplicate using NSCs that were validated by quantitative polymerase chain reaction (qPCR) (Appendix Fig. S1C). The quality of the PHF2 ChIP-seq experiments (Appendix Table S2) was evaluated by analyzing the clustering between samples by Pearson correlation (Appendix Fig. S1D). Moreover, IGV genome browser screenshots in Appendix Fig. S1E illustrate the continuous quantifications of PHF2 ChIP.seq samples within specific regions. We then analyzed PHF2 binding patterns within repetitive regions included in the TEtranscripts (Jin et al, 2015) annotation, which contains more than 3.7 million repeats and it is organized into three levels: Class, Family, and Repeat group. To validate the heterochromatic nature of the classes of repeats of interest, we performed ChIP-seq for H3K9me3 and H3K4me3 histone marks in NSCs and compared them to PHF2 ChIP-seq. As expected, the sequences bound by PHF2 according to Fig. 1C were depleted for H3K4me3 and

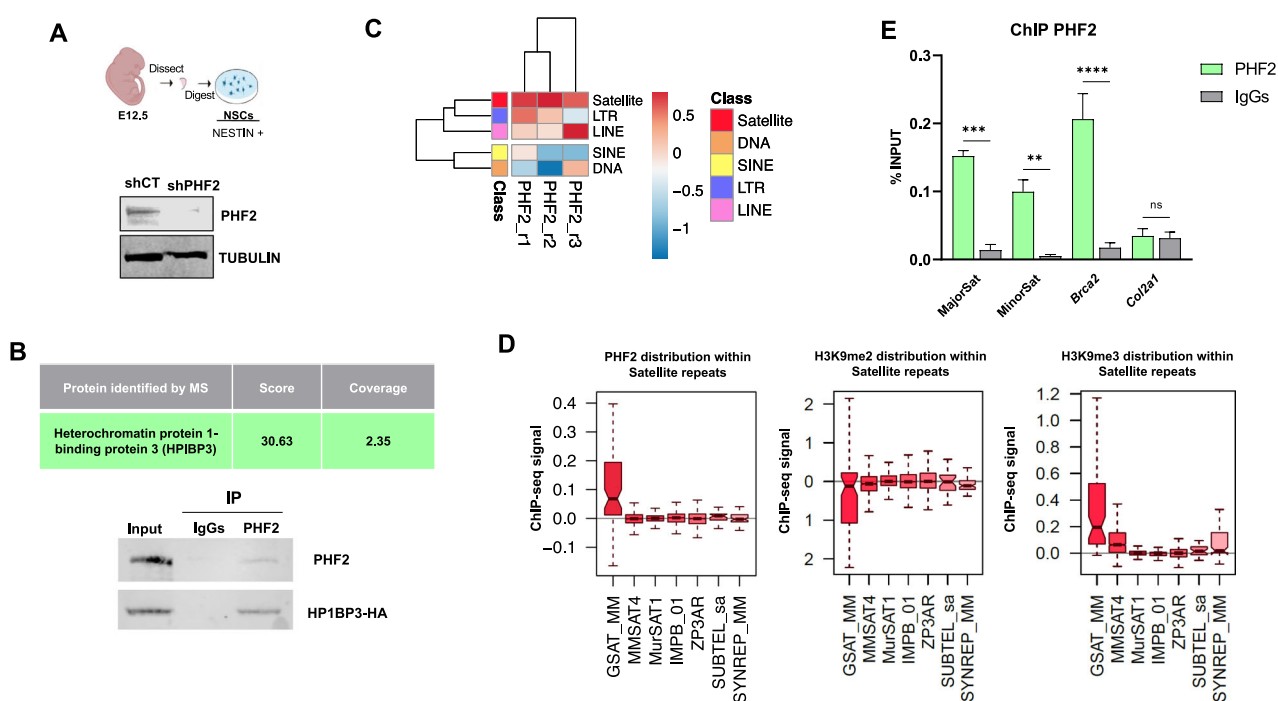

**Figure 1. PHF2 is enriched at satellite repeats.**

(A) Schematic view of the model used in this study. NSCs were dissected from cerebral cortices of mouse fetal brains (E12.5) and cultured ex vivo (see Methods). NSCs were infected with lentivirus expressing shRNA control (shCT) or shRNA specific for PHF2 (shPHF2). Twenty-four hours later shRNA expressing cells were selected with puromycin. Forty-eight hours after infection, total protein extracts were prepared and the PHF2 and TUBULIN levels were determined by immunoblot. The image is representative of two independent experiments with similar results. (B) Endogenous PHF2 protein was immunoprecipitated from shCT or shPHF2 NSCs, using IgGs as negative control, and MS analysis of the immunopellets was performed. HP1BP3 was identified as a possible PHF2's partner. The score and coverage are shown in the table (top). HEK293T cells were transfected with PHF2 and HP1BP3-HA expressing vector. PHF2 was immunoprecipitated using PHF2 antibody and the presence of HP1BP3 in the immunopellet was determined by immunoblot using HA antibody. IgGs were used as negative control. The images are representatives of three independent experiments with similar results. (C) Heatmap and clustering of the mean input-subtracted ChIP-seq signal of three independent PHF2 replicates in Satellite, DNA, SINE, LTR, and LINE classes of repeats. (D) Boxplot showing the PHF2, H3K9me2, and H3K9me3 input-subtracted ChIP-seq signal within seven groups of repeats belonging to Satellite class in shCT condition. The boxplots represent the distribution of the different samples' CPM-normalized and input-subtracted ChIP-seq signal (y-axis) in the corresponding group of repeats. In the case of PHF2, H3K9me2 and H3K9me3, data used for this analysis correspond to the averaged values obtained from three, two and two independent ChIP-seq replicates, respectively. Box plots: centerlines show the medians; box limits indicate the 25th and 75th percentiles; whiskers extend to the minimum and maximum. Exact sample sizes for each group: GSAT_MM = 77; MMSAT4 = 1597; MurSAT1 = 5398; IMPB_01 = 26485; ZP3AR = 2570; SUBTEL_sa = 31; SYNREP_MM = 69. (E) PHF2 ChIP-qPCR assay at major and minor satellite in NSCs. The promoters of Col2a1 and Brca2 genes were used as negative and positive controls, respectively. Data from qPCR were normalized to the input. Results are the mean of three biological independent experiments. Errors bars represent SD. **$p < 0.01$; ***$p < 0.001$ ****$p < 0.0001$ (Student's t-test). Source data are available online for this figure.

presented higher levels of H3K9me3 than other repeats that reside in more euchromatic regions, such as SINEs (Fig. EV1A). Our results indicate that PHF2 binding sites were enriched at specific repetitive elements such as satellites and long interspersed nuclear elements (LINEs) (Fig. 1C). Considering PHF2's reported association with centromeric regions (Ohta et al, 2010), we focused on the presence of PHF2 at various satellite repeats as described by the RepeatMasker tool. The results presented in Fig. 1D and Fig. EV1B,C show a PHF2 enrichment at the major satellite (GSAT_MM) which correlated with low levels of H3K9me2 (the histone mark targeted by PHF2), determined by using already published H3K9me2 ChIP-seq data (Pappa et al, 2019). This also correlated high levels of H3K9me3, as expected for a heterochromatic region. To further confirm the genomic binding or proximity of PHF2 to satellites, we performed PHF2 ChIP followed by qPCR (ChIP-qPCR) targeting major and minor satellite repeats on chromosome 16, one of the best mapped in the mouse genome. Our findings demonstrated significant enrichment of PHF2 at both

major and minor satellites, which was lower than certain TSS previously identified as PHF2 targets; while no enrichment was observed at the Col2a1 promoter region, which was used as a negative control (Fig. 1E).

To broad our conclusion we analyzed the location enrichment of PHF2 at heterochromatic genomic regions in immortalize mouse fibroblast (NIH3T3) by PHF2 ChIP.seq assays, obtaining similar results (Fig. EV1D,E).

Collectively, these results provide compelling evidence that PHF2 is enriched in centromeric satellite repeats, supporting its potential involvement in heterochromatin-associated processes.

## PHF2 maintains heterochromatin silencing

Our previous findings suggest that PHF2 is involved in the regulation of cell division during early neurogenesis (Pappa et al, 2019). Given the importance of heterochromatin stability in cell division and development, we hypothesized that PHF2 may also

play a role in maintaining heterochromatin integrity by counteracting heterochromatin histone-modifying activities and in this way preserving euchromatic modifications. To investigate the role of PHF2 in heterochromatin stability, we performed qPCR assays to assess the expression levels of various heterochromatic repeats in both control and PHF2-depleted NSCs. Our results demonstrated a significant increase in transcription of major and minor satellite repeats upon PHF2 depletion, while the expression of the negative control gene (*Gda*) remained unaffected. A slight increase of expression was also observed in the analyzed interspersed repetitive elements (LINES and SINES), albeit to a lesser extent (Fig. 2A). Similar effects were observed when PHF2 was depleted using a different shRNA (shPHF2_2) (Appendix Fig. S2A). To further understand the impact of PHF2 depletion on heterochromatic repeat expression, we also examined its effects in immortalized mouse embryonic fibroblast NIH3T3 cells. We observed an induction of pericentromeric repeats transcription; however, the overall expression levels were consistently lower compared to those observed in NSCs (Fig. 2B).

To investigate the effect of PHF2 depletion on the organization of PcH, we examined the volume of PcH clusters, which form chromocenters in mouse cells (Jones, 1970; Pardue and Gall, 1970). This was accomplished by measuring the volume of DAPI (DNA stain) and H3K9me3 (marker of PcH) foci in control and PHF2-depleted NSCs (Fig. 2C) and NIH3T3 cells (Fig. 2D). Interestingly, we observed an increase in the volume of chromocenters, suggesting alterations in PcH alteration upon PHF2 depletion (Fig. 2C,D; Appendix Fig. S2B). Taken together, these findings provide further support for the role of PHF2 in maintaining the transcriptional status of pericentromeric satellites, thereby influencing the organization of PcH.

## PHF2 balances H3K9me3 levels

To gain insights into how PHF2 maintains heterochromatin stability, we investigated the impact of PHF2 depletion on H3K9me3 levels. We performed duplicated H3K9me3 ChIP-seq analysis in shCT and shPHF2 NSCs (Appendix Fig. S3A,B), that were validated by qPCRs assays (Appendix Fig. S3C). Initially, we examined the global changes in chromatin methylation by segmenting the genome into 15 Kb bins. This analysis revealed that PHF2 depletion led to higher H3K9me3 signal strength in 168 bins and lower signal strength in 22 bins compared to control cells (Fig. 3A). Moreover, we observed increased H3K9me3 levels around the PHF2 binding sites (Fig. 3A, bottom panel). These findings align with H3K9me3 immunostaining assays,

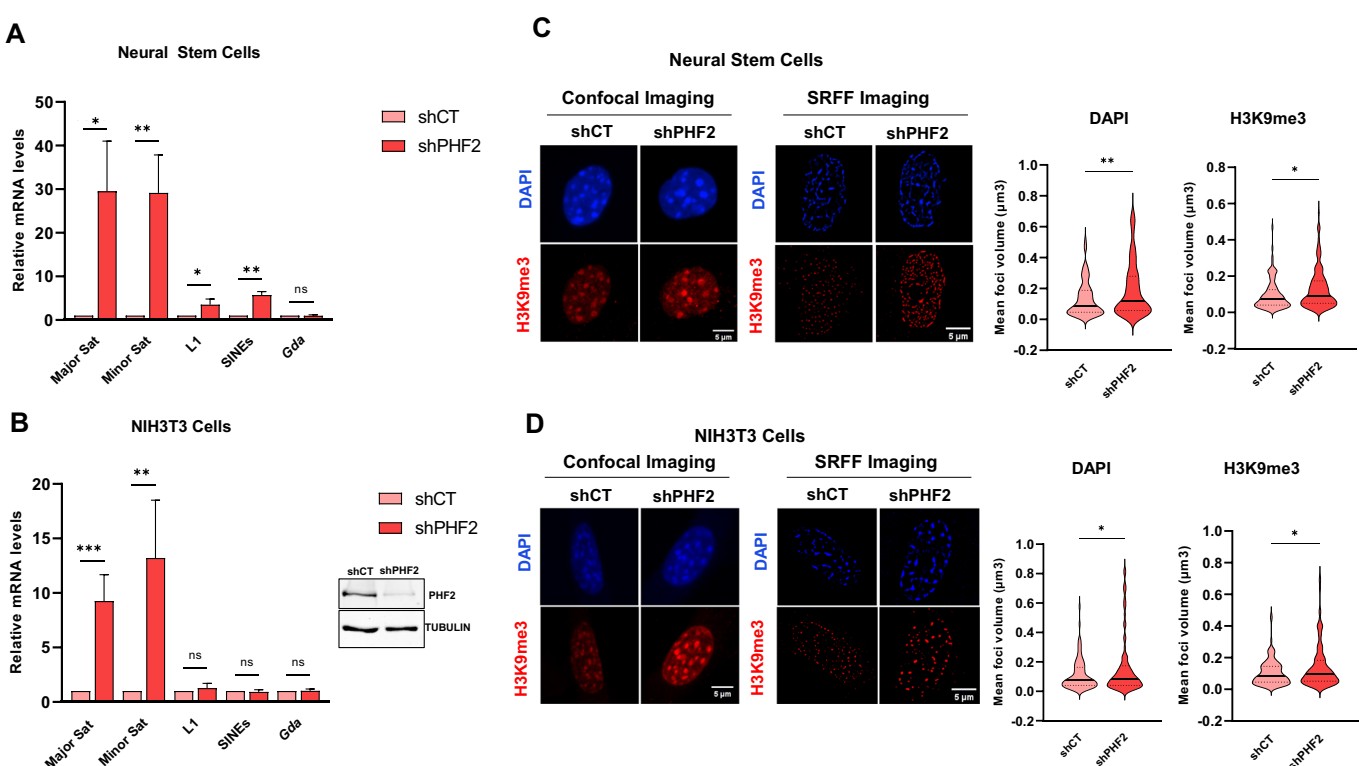

**Figure 2. PHF2 depletion leads to centromeric satellite transcription.**

(**A**) RNA levels of major satellite, minor satellite, L1-Lines and Sines in control (shCT) and PHF2 depleted (shPHF2) NSCs were analyzed by qPCR. Data were normalized to *Gapdh* reference gene levels and figure shows values relative to shCT cells. *Gda* gene was used as a negative control. Error bars indicate SD and $n = 3$. *$p < 0.05$; **$p < 0.01$ (Student's t-test). (**B**) The same than in A, but NIH3T3 cells were used. **$p < 0.01$; ***$p < 0.001$ (Student's t-test). (**C**) Immunostaining assays of shCT and shPHF2 NSCs. Cells were fixed and stained using anti-H3K9me3 antibody and DAPI to visualize the DNA. Confocal and super resolution (SRRF) images were taken (see Methods). The volume of the DAPI and H3K9me3 foci were determined using ImageJ software (see Methods). Data shown are representative of three biologically independent experiments. Scale bar indicates 5 µm. Violin plots represent the foci volume quantification of 100 cells ($n = 100$). *$p < 0.05$; **$p < 0.01$ (Student's t-test). (**D**) The same than in (**C**), but NIH3T3 cells were used. Source data are available online for this figure.

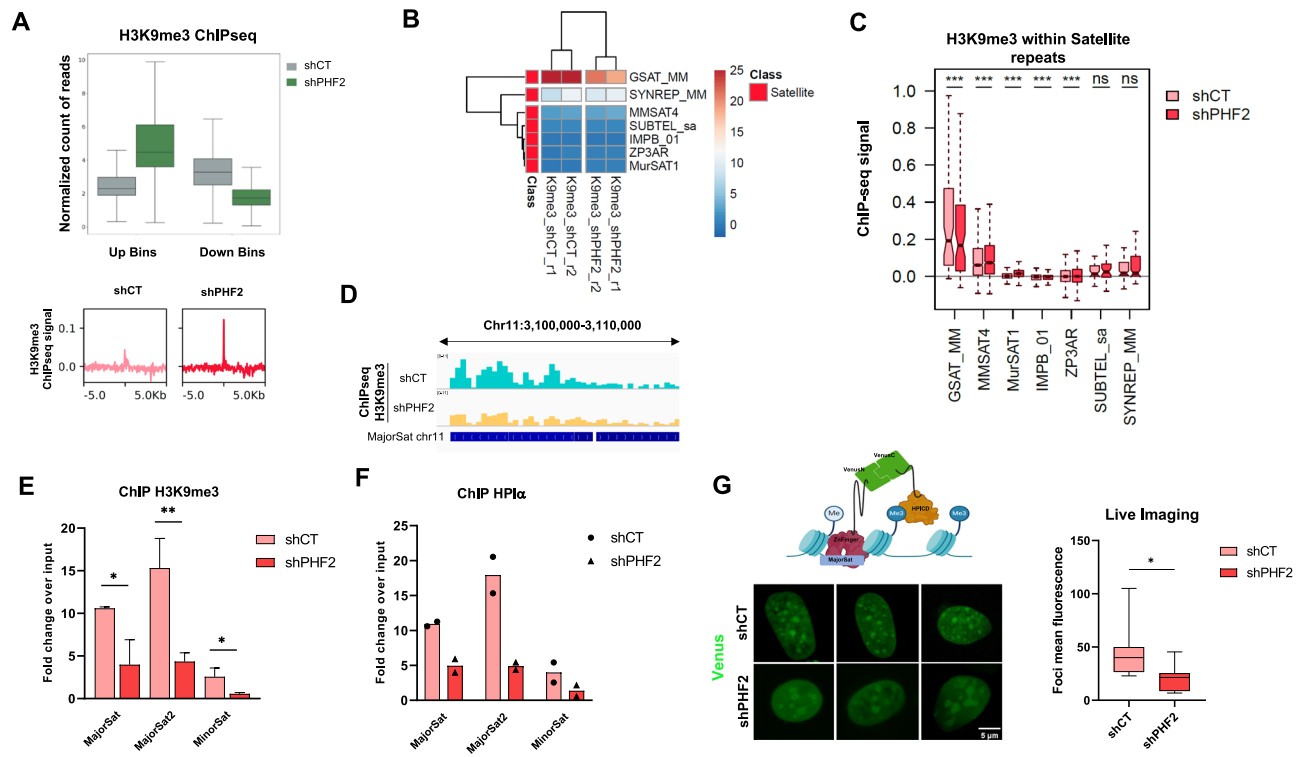

**Figure 3. PHF2 balances H3K9me3.**

(A) H3K9me3 ChIP-seq was performed in duplicate in shCT and shPHF2 NSCs. The mouse genome was segmented into $n = 180{,}000$ 15 Kb bins and analyzed for H3K9me3 enrichment. Boxplots represent the bins that gain and loss H3K9me3 and shCT and shPHF2 bins appear separated (top part of the figure). H3K9me3 input-subtracted ChIP-seq enrichment in shCT vs shPHF2 NSCs at PHF2 binding sites $+/5000$ bp determined by ChIP-seq (bottom part of the figure). Box plots: centerlines show the medians; box limits indicate the 25th and 75th percentiles; whiskers extend to the minimum and maximum. (B) Heatmap and clustering of the mean input-subtracted ChIP-seq signal of H3K9me3 within seven groups of repeats belonging to Satellite class. Two independent replicates of H3K9me3 in shCT and shPHF2 cells are shown. (C) Boxplot showing the H3K9me3 input-subtracted ChIP-seq signal within seven groups of repeats belonging to Satellite class in shCT and shPHF2 cells. The boxplots represent the distribution of the different samples' CPM-normalized and input-subtracted ChIP-seq signal (y-axis) in the corresponding group of repeats. Data used for this analysis correspond to the averaged values obtained from two independent ChIP-seq replicates. ***$p < 0.001$; (Wilcoxon signed-rank test). Box plots: centerlines show the medians; box limits indicate the 25th and 75th percentiles; whiskers extend to the minimum and maximum. Exact sample sizes for each group: GSAT_MM = 77; MMSAT4 = 1597; MurSAT1 = 5398; IMPB_01 = 26485; ZP3AR = 2570; SUBTEL_sa = 31; SYNREP_MM = 69. (D) Integrated Genome Viewer (IGV) captures showing input-substracted H3K9me3 ChIP-seq levels at major satellite of chromosome 11 in control (shCT) and PHF2 depleted (shPHF2) NSCs. Scale [0–11]. (E) H3K9me3 ChIPs in shCT and shPHF2 NSCs were analyzed by qPCR at the major and minor satellites. Data from qPCR were normalized to the input and the IgG values subtracted. Results are the mean of three biological independent experiments. Errors bars represent SD. *$p > 0.05$; **$p < 0.01$ (Student's t-test). (F) HP1α ChIPs were performed in shCT and shPHF2 NSCs and qPCR analysis was carried out at at the major and minor satellites. Data from qPCR were normalized to the input the IgG values subtracted. Results are the mean of two biological independent experiments. (G) Diagram illustrating the BiAD Venus-based sensor configuration. The HP1 chromodomain (HP1CD) interacts with H3K9me3, while the zinc-finger domain binds to mouse major satellite DNA repeats. The close proximity of the two Venus subunits (C-terminal, VenusC; N-terminal, VenusN) results in the emission of a fluorescent signal (upper panel). Confocal live-cell imaging analysis was conducted on shCT and shPHF2 NIH3T3 cells expressing the BiAD Venus-based sensor (displaying a green signal) for the detection of H3K9me3 at major satellite repeats (Lungu et al, 2017). Scale bar indicates 5 μm (bottom panel). Boxplot represents the mean foci fluorescence intensity in shCT and shPHF2 NIH3T3 cells from 3 biologically independent experiments. Box plots: centerlines show the medians; box limits indicate the 25th and 75th percentiles; whiskers extend to the minimum and maximum. Error bars represent SD. *$p < 0.05$ (Student's t-test). Source data are available online for this figure.

demonstrating an overall increase in bulk H3K9me3 methylation upon PHF2 depletion (Fig. 2C,D and (Pappa et al, 2019)). Interestingly, the bins that showed an increase in H3K9me3 levels upon PHF2 depletion were found to be enriched at transcription start sites (TSS) (Appendix Fig. S3D). Furthermore, gene ontology (GO) analysis of these regions revealed their association with cell cycle, DNA transcription, and other fundamental processes related to cell proliferation and gene expression (Appendix Fig. S3E). These findings reinforce the notion that PHF2 maintains chromatin accessibility at regions that are crucial for cell proliferation, as previously suggested (Pappa et al, 2019). However, the situation was different when we examined the satellite repeats, particularly

the major satellite. In this case, a decrease in H3K9me3 levels was observed (Fig. 3B–D), data that was confirmed by ChIP-qPCR (Fig. 3E), and correlated with a decrease in HP1α binding (Fig. 3F) providing a molecular explanation for the observed activation of PcH transcription. To prove that epigenetic instability was associated with a H3K9me3 decrease at heterochromatin regions containing major satellites, we utilized modular fluorescence complementation sensors. These sensors allowed us to detect H3K9me3 signals specifically at the major satellite genomic site in live cells, as illustrated in Fig. 3G (Lungu et al, 2017; Panatta et al, 2022). Live-cell imaging of both control and PHF2-depleted cells revealed a decrease in H3K9me3 levels at major satellite regions

compared to the remaining cellular levels. Furthermore, we observed that methylated foci exhibited a higher volume in PHF2-depleted cells compared to control cells, consistent with the data presented in Fig. 2C,D.

The boundaries are important regions in maintaining the stability of heterochromatin, so after observing a loss in methylation in the repeats, we decided to analyze the H3K9me3 levels near them. For this analysis, we divided the mouse genome into 10 Kb bins and examined the changes in H3K9me3 levels across the 5 megabse (Mb) region from the centromeric end of each mouse chromosome in control and PHF2-depleted samples. Interestingly, in contrast to what was observed at the satellite repeats (Fig. 3B–E), we found a significant increase of H3K9me3 levels, mainly within the first 500 Kb next to the chromosome centromeres (3–3.5 Mb) upon PHF2 depletion (Fig. 4A,B), that was confirmed by ChIP-qPCR assays (Fig. 4C). Since H3K9me3 is recognized by HP1α, which facilitates silencing through a read-write mechanism (Grewal, 2023; Martens et al, 2005), we decided to analyze the binding of HP1α in these regions using ChIP-qPCR. The

results indicate that, in accordance with the increase in H3K9me3 levels, the binding of HP1α was also increased (Fig. 4D). In addition, no significant changes on H3K4me3 levels were observed upon PHF2 depletion either in the vicinity of PcH (Fig. EV2A) or at global level (Fig. EV2B).

Our findings reveal that the depletion of PHF2 triggers an increase in H3K9me3 within genomic boundaries (Fig. 4A,B) and other genomic regions, specially TSS (Pappa et al and Fig. 3A) potentially leading to destabilize heterochromatin, particularly if the availability of heterochromatin components is limited, thus facilitating increased transcription of repetitive elements. To investigate this hypothesis, we focused on specific genes marked by H3K9me3 in control NSCs, which were subsequently found to be upregulated upon PHF2 depletion. To ascertain whether this increased transcription following PHF2 depletion is mediated indirectly through the redistribution of heterochromatin components, we turned our attention to HP1α. ChIP-qPCR analysis of HP1α revealed a decrease in its recruitment to analyzed

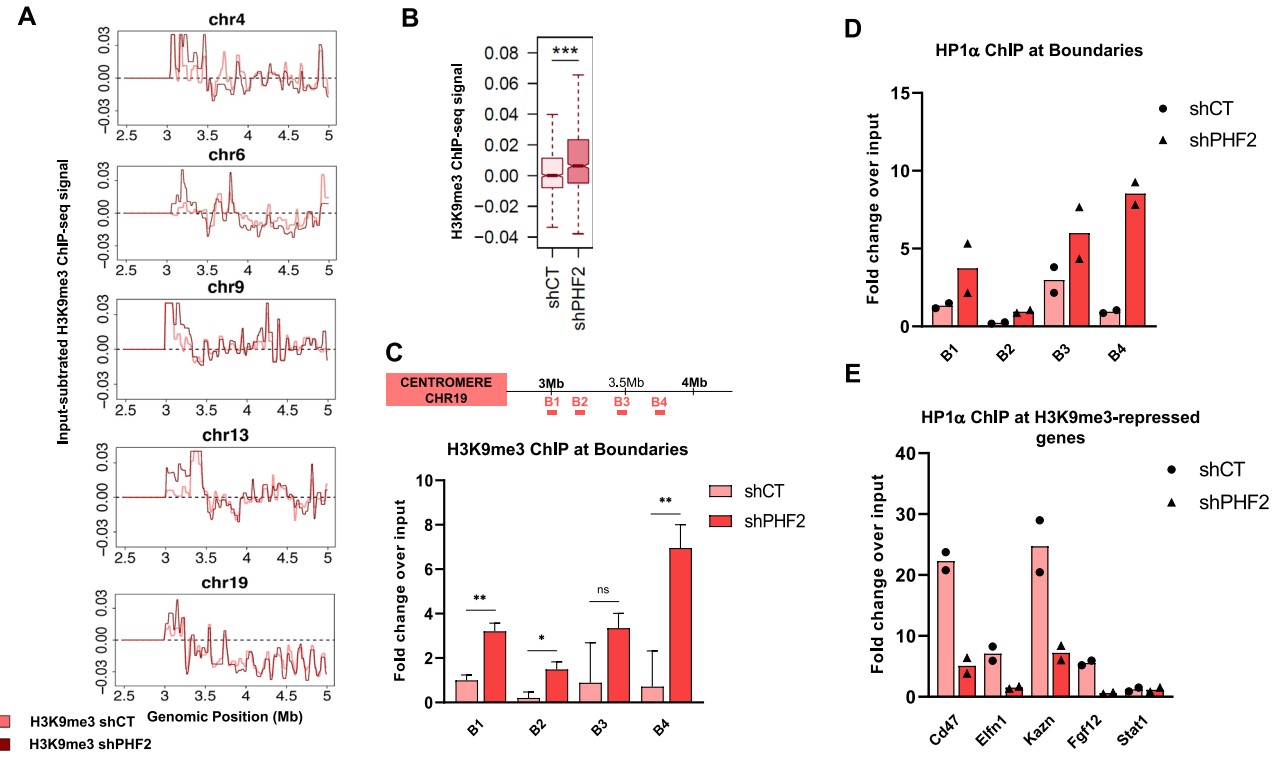

**Figure 4. PHF2 maintains H3K9me3 at boundaries.**

(A) Line plot showing the H3K9me3 profile across the first 5 Mb (2.5–5 Mb) from the centromeric end of chromosomes 4, 6, 9, 13, and 19 in both shCT and shPHF2 conditions. Data used for this analysis correspond to the averaged input-subtracted H3K9me3 ChIP-seq signal values obtained from two independent replicates. (B) Boxplot displaying the H3K9me3 input-subtracted ChIP-seq signal within 1000 pericentromeric bins (3–3.5 Mb) from all chromosomes in shCT and shPHF2 cells. The boxplots represent the distribution of the different samples' CPM-normalized and input-subtracted ChIP-seq signal (y-axis) in the corresponding group of bins (x-axis). Data used for this analysis correspond to the averaged values obtained from two independent H3K9me3 ChIP-seq replicates. Box plots: centerlines show the medians; box limits indicate the 25th and 75th percentiles; whiskers extend to the minimum and maximum. ***$p < 0.001$ (Wilcoxon signed-rank test). Exact sample sizes for each group (the same pericentromeric bins were considered in both cases to evaluate changes triggered by the PHF2 KD in these regions): shCT = 1000; shPHF2 = 1000. (C) H3K9me3 ChIPs in shCT and shPHF2 NSCs were analyzed by qPCR within the genomic region spanning positions B1-B4, situated between 3–4 Mb from the centromere of chromosome 19 (see top panel). Data from qPCR were normalized to the input and the IgG values subtracted. Results are the mean of three biological independent experiments, each performed with technical triplicates. Errors bars represent SD. *$p < 0.05$; **$p < 0.01$ (Student's t-test). (D, E) HP1α ChIPs were performed in shCT and shPHF2 NSCs. In (D), qPCR analysis was carried out at positions B1-B4, located between 3–4 Mb from the centromere of chromosome 19. In (E), qPCR was conducted at specific genes as indicated. Similar to H3K9me3 analysis, qPCR data were normalized to the input and the IgG values subtracted. Results represent the mean of two independent biological experiments. Source data are available online for this figure.

heterochromatic loci upon PHF2 depletion, while an increase was observed over the boundary regions (Fig. 4E). This observation suggests a potential reorganization of HP1α, and likely other heterochromatin components, upon PHF2 depletion.

## PHF2 is enriched at the PcH boundaries

After observing changes on the H3K9me3 pattern around the PcH, we decided to reanalyze the PHF2 ChIP-seq searching for the presence of PHF2 at those regions. Interestingly, we observed an enrichment of PHF2 in a 5 Mb region from the centromeric end of

each mouse chromosome, particularly significant within the first 500 Kb next to the centromere (3–3.5 Mb) (Fig. 5A,B). Indeed, the ChIP-seq signal within 1000 bins corresponding to these initial pericentromeric regions (3–3.5 Mb) of mouse chromosomes was significantly higher than within 1000 random genomic windows (Fig. 5C). The enrichment of PHF2 in this region was confirmed by ChIP-qPCR assays (Appendix Fig. S4A) and through PHF2 immunostaining assays and super-resolution microscopy, where it was found to be in close proximity to, but not colocalized with, chromocenters (Fig. 5D). Notably, PHF2 was found to be closer to the chromocenter and nuclear lamina compared to other members

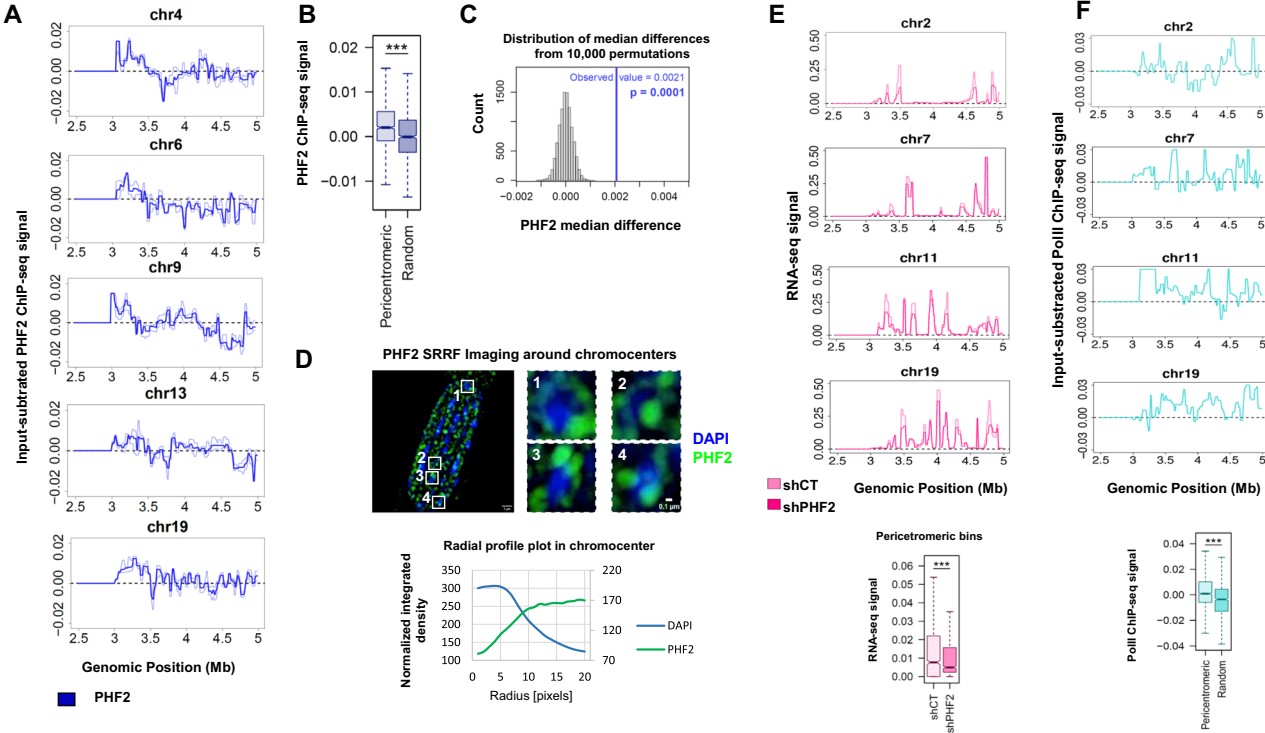

**Figure 5. PHF2 binds PcH boundaries.**

(**A**) Line plot showing the PHF2 profile across the first 5 Mb (2.5–5 Mb) from the centromeric end of chromosomes 4, 6, 9, 13, and 19. Data used for this analysis correspond to the averaged input-subtracted values within 10 Kb bins obtained from three independent PHF2 ChIP-seq replicates. Lighter lines denote SE. (**B**) Boxplot displaying the PHF2 input-subtracted ChIP-seq signal within 1000 pericentromeric bins (3–3.5 Mb) from all chromosomes compared to 1000 random bins in shCT condition. The boxplots represent the distribution of the different samples' CPM-normalized and input-subtracted ChIP-seq signal (y-axis) in the corresponding group of bins (x-axis). Box plots: centerlines show the medians; box limits indicate the 25th and 75th percentiles; whiskers extend to the minimum and maximum. Data used for this analysis correspond to the averaged values obtained from three independent PHF2 ChIP-seq replicates. ***$p < 0.001$ (Mann–Whitney U test). (**C**) Histogram of the median differences from the permutation test between PHF2 ChIP-seq quantification in pericentromeric and random bins. A total of $N = 10,000$ permutations were performed. The blue vertical bar corresponds to the observed difference in medians between PHF2 ChIP-seq signal in pericentromeric and random bins. (**D**) Super-resolution (SRRF) imaging of PHF2 (green) and DNA (blue) on NIH3T3 cells. Left image shows the entire nucleus, and captures 1–4 show different DAPI foci with higher amplification and resolution and PHF2 surrounding them (top panel). Radial profile plot of DAPI (blue) and PHF2 (green) intensities on a ROI defined around nuclear foci in NIH3T3 cells (bottom panel). (**E**) Line plot depicting the average quantification of RNA-seq data in shCT and shPHF2 NSCs within 10 Kb intervals along the first 5 Mb of chromosomes 2, 7, 11, and 19. The analysis is based on RNA-seq in NSCs data derived from two biological independent replicates (upper panel). Boxplot comparing the average RNA-seq quantification within 1000 pericentromeric bins (3–3.5 Mb) from all chromosomes to 1000 random bins under shCT and shPHF2 conditions. The boxplot represents the distribution of the different samples' CPM-normalized RNA-seq signal (y-axis) across the corresponding group of bins (x-axis). Box plots: centerlines show the medians; box limits indicate the 25th and 75th percentiles; whiskers extend to the minimum and maximum. Statistical differences between shCT and shPHF2 RNA-seq quantification at the same genomic bins (pericentromeric) were evaluated using the Wilcoxon signed-rank test (***$p < 0.001$). Exact sample sizes for each group (the same pericentromeric bins were considered in both cases to evaluate changes triggered by the PHF2 KD in these regions): shCT = 1000; shPHF2 = 1000 (lower panel). (**F**) Line plot showcasing the RNA PolII (GSM1635080) profile across the first 5 Mb (2.5–5 Mb) from the centromeric end of chromosomes 2, 7, 11, and 19 in NSCs. Data used for this analysis correspond to the averaged input-subtracted ChIP-seq values within 10 Kb bins (upper panel). Boxplot displaying the RNA PolII input-subtracted ChIP-seq signal within 1000 pericentromeric bins (3–3.5 Mb) from all chromosomes compared to 1000 random bins. The boxplots represent the distribution of the different samples' CPM-normalized and input-subtracted ChIP-seq signal (y-axis) in the corresponding group of bins (x-axis). Box plots: centerlines show the medians; box limits indicate the 25th and 75th percentiles; whiskers extend to the minimum and maximum. ***$p < 0.001$ (Mann–Whitney U test) (lower panel). Exact sample sizes for each group: Pericentromeric = 1000; Random = 1000. Source data are available online for this figure.

of the HDM7 family, including PHF8 and KIAA1718, used as a control (Appendix Fig. S4B).

Given the described link between chromatin dynamics at heterochromatin boundaries and heterochromatin stability in organisms ranging from yeast to mammals (Donze and Kamakaka, 2001; Ebersole et al, 2011; Lunyak et al, 2007; Raab et al, 2012; Scott et al, 2006) we examined the impact of PHF2 depletion on transcription within these regions. To do that we reanalyzed previously published RNA-seq data in NSCs (Pappa et al, 2019). Our analysis revealed a decrease in transcription levels in the pericentromeric regions upon PHF2 depletion (Fig. 5E) suggesting that PHF2 directly or indirectly contributes to maintaining the transcriptional activity in these regions. Interestingly, these regions exhibit a higher level of RNA PolII binding compared to randomly chosen regions (Fig. 5F), along with a lower level of histone H1 binding (Fig. EV3A), suggesting that they are transcriptionally more active than the genome-wide average.

Thus, the transcriptional decrease observed upon PHF2 depletion (Fig. 5E), as well as the observed redistribution of heterochromatin components (HP1α) (Fig. 4E), could potentially contribute to the instability of pericentromeric repeats.

To gain insight into the nature of the boundary regions, we conducted a comprehensive characterization of the 3–5 Mb downstream of the repeats. Initial analyses revealed that these regions lack significant G/C features. On the other hand, it is well established that genes for noncoding transfer RNAs (tRNAs) function as boundary elements (Donze and Kamakaka, 2001; Ebersole et al, 2011; Raab et al, 2012; Scott et al, 2006). Therefore, we investigated the presence of tRNAs and ncRNAs and observed an enrichment of these genomic elements across the boundary regions (Fig. EV3B,C). However, we have been unable to assess the potential role of PHF2 in their expression due to our inability to detect them in the RNA-seq data.

Collectively, the results suggest that PHF2 plays a crucial role in balancing H3K9me3 levels. This is essential for stabilizing PcH either by probably regulating transcription at the boundaries, and/or disrupting the distribution of heterochromatin components.

## PHF2 regulates chromatin accessibility

To deeply understand how PHF2 depletion leads to heterochromatin instability, we conducted ATAC-seq assays in both control and two replicates of depleted PHF2 NSCs (Appendix Fig. S5A). These assays allow us to assess chromatin accessibility and provide insights into the changes associated with PHF2 function. The results revealed that the number of regions exhibiting decreased accessibility upon PHF2 depletion was slightly higher than those showing increased accessibility (Fig. 6A). This observation is consistent with the role of PHF2 as an H3K9me2 demethylase. Further analysis demonstrated that the regions that lost accessibility were significantly enriched in promoters, particularly at transcription start sites (TSS) (Appendix Fig. S5B-D), and were associated with genes involved in cell cycle progression, DNA transcription, and other fundamental processes related to cell proliferation (Appendix Fig. S5E). These findings align with the known role of PHF2 in NSCs (Pappa et al, 2019). On the other hand, the regions that gained chromatin accessibility were primarily located in intergenic regions (Appendix Fig. S5C). Interestingly, regions occupied by PHF2 lost accessibility when

PHF2 was depleted (Fig. 6B). These data strongly suggest that PHF2 is responsible for maintaining chromatin accessibility at these specific sites, corroborating previous findings from different experiments (Pappa et al, 2019). Keeping in mind the potential role of PHF2 in heterochromatin destabilization, we focused on the PcH. Specifically, we examined the global changes in chromatin accessibility at the satellite repeats. Intriguingly, our analysis revealed a significant increase in chromatin accessibility at these repeats, as depicted in Fig. 6C–E.

Next, we examined the changes on chromatin accessibility across the boundary regions of interest in control and PHF2-depleted samples. Interestingly, in contrast to what was observed at the satellite repeats (Fig. 6C,D), we found a slight but significant decrease of ATAC signal, next to the chromosome centromeres, upon PHF2 depletion (Fig. 6F,G). This decrease is consistent with the enhanced H3K9me3 levels observed over this region (Fig. 4A–C).

These data suggest that PHF2 keeps chromatin accessible at promoters in which it is located. However, at satellite repeats, depletion of PHF2 resulted in an increase in chromatin accessibility.

## The maintenance of PcH stability relies on the PHD and JmjC domains within PHF2

In order to gain insights into the mechanism underlying PHF2-mediated heterochromatin silencing, we aimed to identify the specific domains of PHF2 involved in the observed unscheduled transcription from PcH. PHF2 comprises two well-established protein domains: the PHD domain, which is known for its H3K4me3-binding activity, and the JmjC domain, which possesses demethylase activity (Fueyo et al, 2015). In addition, a recently described charged region in the IDR has been described to be involved in nuclear condensate formation, playing an important role in transcription control (Vicioso-Mantis et al, 2022). To assess the contribution of these domains to the transcription of heterochromatic repeats, we initially confirmed their comparable expression in NIH3T3 cells (Fig. 7A). Subsequently, we performed rescue experiments by introducing either PHF2 wild type (WT) or specific PHF2 mutants into either NSCs or NIH3T3 cells depleted of PHF2, and assessed the expression of the satellite sequences. As depicted in Fig. 7B and Fig. EV4A, both PHF2 WT and the charged mutant in the IDR (ΔCharged) effectively rescued the unscheduled satellite expression. In contrast, the catalytic and PHD domains were unable to rescue the transcription, and instead, they exacerbated the aberrant PcH transcription (Figs. 7B and EV4A). These findings were further supported by Fig. EV4B, where overexpression of the PHD (ΔPHD) and catalytic (HD/AA) PHF2 mutants induced satellite transcription, unlike the PHF2 WT or ΔCharged mutant. Furthermore, and accordingly with the results in Fig. 2C,D, both PHF2 WT and the ΔCharged mutant successfully reversed the increased volume of H3K9me3 foci observed in PHF2 depleted cells. In contrast, the catalytic and PHD domains were unable to achieve this effect (Fig. EV4C). Collectively, these findings provide evidence supporting the role of PHF2 catalytic activity (probably maintaining the H3K9me3 balance), and the PHD domain (likely facilitating the recognition of euchromatin histone marks) in maintaining the transcriptional status of pericentromeric satellites.

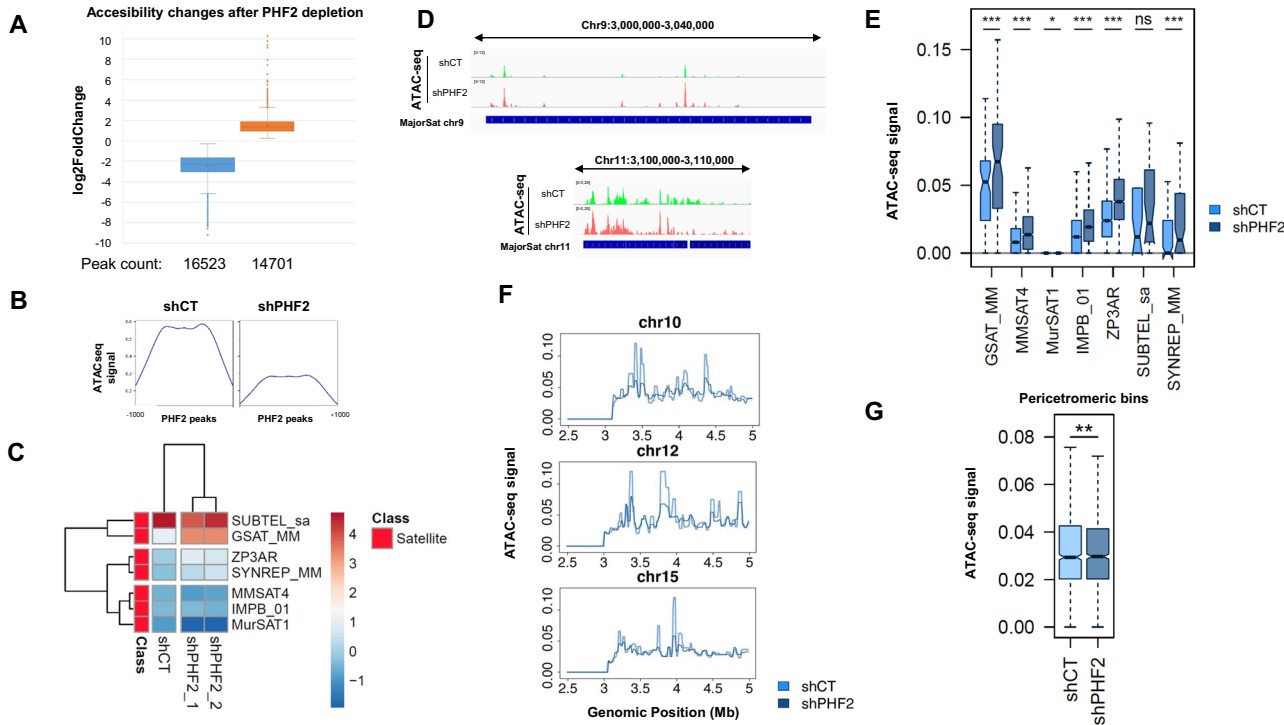

**Figure 6. PHF2 regulates chromatin accessibility.**

(A) ATAC-seq assay was performed in shCT and shPHF2 NSCs. Peak calling was done as described in Methods and fold change calculated. Boxplots represent the peaks that gain (orange box) and loss (blue box) ATAC signal. Total n = 31224 peaks. Box plots: centerlines show the medians; box limits indicate the 25th and 75th percentiles; whiskers extend to the minimum and maximum. Error bars represent SD. (B) ATAC-seq read distribution at PHF2 binding sites +−1000 bp in shCT and shPHF2 NSCs. (C) Heatmap and clustering of the mean input-subtracted ATAC-seq signal within seven groups of repeats belonging to Satellite class. (D) Integrated Genome Viewer (IGV) captures showing ATAC signals at major satellite of chromosome 11and 9 in shCT and shPHF2 NSCs. Scale [0–12]. (E) Boxplot showing the ATAC-seq signal within seven groups of repeats belonging to Satellite class in shCT and shPHF2 NSCs. The boxplots represent the distribution of the different samples' CPM-normalized ATAC-Seq signal (y-axis) in the corresponding group of repeats. Box plots: centerlines show the medians; box limits indicate the 25th and 75th percentiles; whiskers extend to the minimum and maximum. Data used for this analysis correspond to the averaged values obtained from two independent ATAC-seq replicates. *p < 0.01; ***p < 0.001 (Wilcoxon signed-rank test). Exact sample sizes for each group: GSAT_MM = 77; MMSAT4 = 1597; MurSAT1 = 5398; IMPB_01 = 26485; ZP3AR = 2570; SUBTEL_sa = 31; SYNREP_MM = 69. (F) Line plot showing the accessibility profile across the first 5 Mb (2.5–5 Mb) from the centromeric end of chromosomes 10, 12, and 15 (ATAC-seq signal) in both shCT and shPHF2 conditions. Data used for this analysis correspond to the averaged values obtained from two independent ATAC-seq replicates. (G) Boxplot displaying the input-subtracted ATAC-seq signal within 1000 pericentromeric bins (3–3.5 Mb) from all chromosomes in shCT and shPHF2 cells. Data used for this analysis correspond to the averaged values obtained from two independent ATAC-seq replicates. Box plots: centerlines show the medians; box limits indicate the 25th and 75th percentiles; whiskers extend to the minimum and maximum. **p < 0.01 (Wilcoxon signed-rank test). Exact sample sizes for each group (the same pericentromeric bins were considered in both cases to evaluate changes triggered by the PHF2 KD in these regions): shCT = 1000; shPHF2 = 1000. Source data are available online for this figure.

## Unscheduled PHF2-induced heterochromatin transcription results in DNA damage

PHF2 depletion has been linked to accumulation of double-strand breaks (DSBs) and R-loops (Pappa et al, 2019). DNA damage caused by unscheduled transcription from PcH is an important cause of this phenomenon. Therefore, we decided to analyze the direct relationship between repeat transcription, DSBs, and genome instability. To accomplish this, we utilized the described PHF2 mutants as some of them were capable of rescuing unscheduled repeat transcription while others were not. Thus, we evaluated their ability to rescue DNA damage induced by PHF2 depletion, as detected through immunostaining with γH2Ax. The results from Fig. 7C demonstrated that Δ Charged mutant, known for its importance in gene transcriptional activity (Vicioso-Mantis et al, 2022), completely rescued DNA breaks, similar to PHF2 WT. However, the PHD and the catalytic mutants which induce repeat expression, failed to do so. This observation was

further supported by Fig. EV4D, showing that only overexpression of the PHD and catalytic PHF2 mutants, but not the WT, resulted in DNA damage. On the other hand, the genomic instability generated upon PHF2 depletion (Pappa et al, 2019) was efficiently rescued by PHF2 WT and Δ Charged mutant, but not by the PHF2 PHD or catalytic domain mutants (Fig. 7D).

These data suggest that besides its well-characterized function at the gene promoter, PHF2 also plays a crucial role in maintaining PcH stability through its PHD and catalytic domains. This newly described role of PHF2 safeguards genome instability, thereby emphasizing the multifaceted nature of PHF2's functions.

## Discussion

Understanding the role of histone modifications in maintaining genome stability during neural development and their contribution

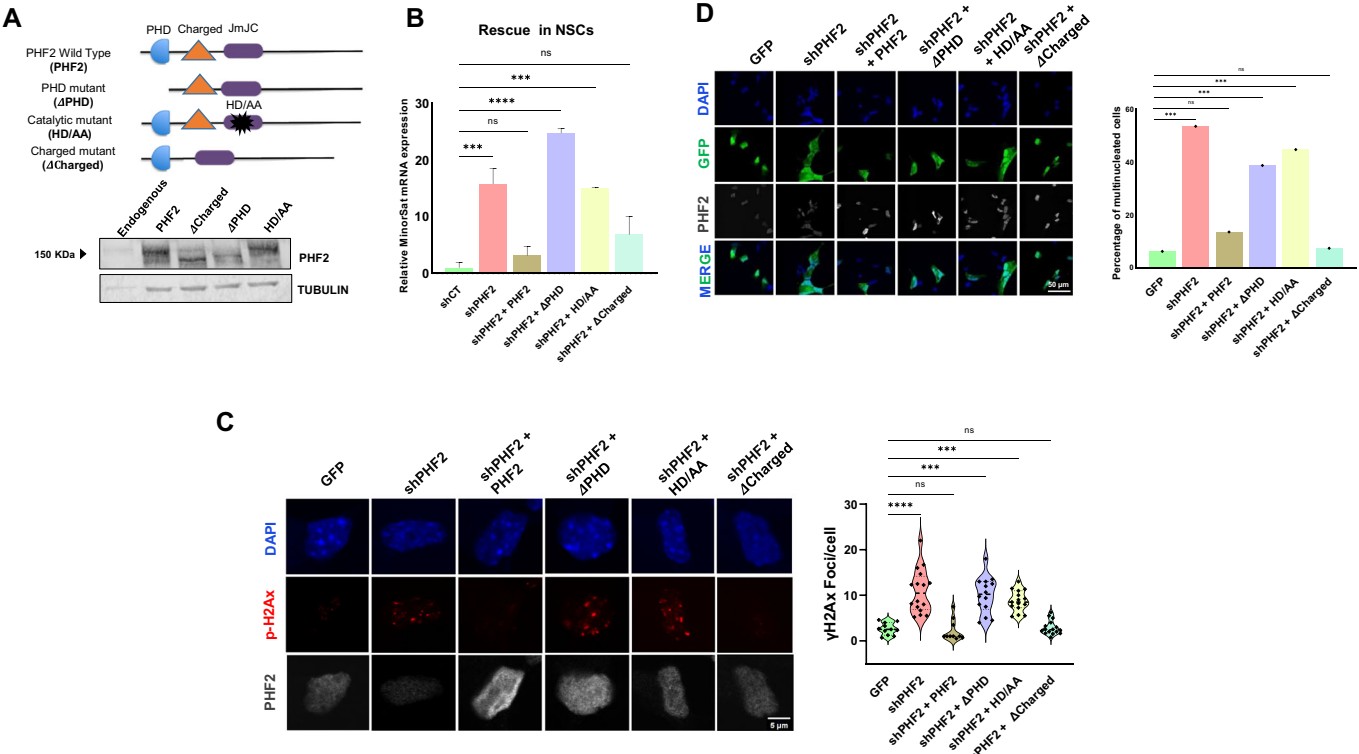

**Figure 7. The maintenance of PcH stability relies on the PHD and JmjC domains within PHF2.**

(A) Immunoblot showing PHF2 mutants expression levels using anti-PHF2 and TUBULIN antibodies. The images are representatives of two independent experiments with similar results. (B) PHF2-depleted NSCs were infected with lentivirus expressing PHF2 WT, ΔCharged, ΔPHD, or HD/AA mutants. Total RNA was prepared and the levels of RNA levels of minor satellite, was determined by qPCR. Mean was calculated from data of 3 biologically independent experiments and normalized to *Gapdh* reference gene levels and figure shows values relative to shCT cells. Error bars indicate SD. ***$p < 0.001$, ****$p < 0.0001$ (Student's t-test). (C) shPHF2 NSCs were nucleofected either with GFP alone or with PHF2 WT, ΔCharged, ΔPHD, or HD/AA mutants. Cells were fixed and stained with PHF2 and γH2Ax antibodies and DAPI. Green cells were analyzed. More than 30 cells were quantified. Data shown are representative of 3 biologically independent experiments. Scale bar indicates 5 μm. Violin plots represent the number of γH2Ax foci/cell. ***$p < 0.001$, ****$p < 0.0001$ (Student's t-test). (D) The same than (C) but the cells were stained with DAPI and the % of multinucleated cells was determined. Diagram depicts the % of multinucleated cells in each condition. More than 100 cells were quantified. Data shown are representative of 3 biologically independent experiments. Scale bar indicates 5 μm. ***$p < 0.001$ (Student's t-test). Source data are available online for this figure.

to disease pathology is a significant challenge in the field. Among these modifications, H3K9 methylation is a crucial and conserved mark involved in chromatin silencing. Based on the findings presented in this study, we propose that PHF2-mediated H3K9 demethylation, in addition to its role in transcriptional control, plays a pivotal function in stabilizing PcH repetitive sequences. This is crucial for preventing the accumulation of DNA damage and maintaining genome stability, which are essential for the expansion of neural progenitor cells.

## PHF2 as a regulator of heterochromatin stability

We have demonstrated that PHF2 plays a catalytic-dependent role in preventing repeat transcription. Our findings indicate that maintaining a precise balance of H3K9me3 throughout the genome is essential for the stability of repeats. An increase of this mark in boundary regions, as well as other genomic areas, could potentially promote abnormal repeat transcription, likely via dilution or redistribution of heterochromatin components, as illustrated by our findings with HP1α. On the other hand, PHF2 depletion correlates with decreased transcription levels at the boundaries. Studies have

elucidated the interplay between transcription and RNA-mediated mechanisms functioning as boundary elements in organisms spanning from yeast to mammals (Donze and Kamakaka, 2001; Ebersole et al, 2011; Lunyak et al, 2007; Raab et al, 2012; Scott et al, 2006). Therefore, it's plausible that PHF2, a well-characterized transcriptional regulator, could modulate PcH stability by influencing local RNA levels at PcH boundaries. Furthermore, besides the potential impact of RNA transcripts, the PHF2-mediated transcription process likely induces changes in histone modifications or nuclear positioning, thus efficiently regulating heterochromatin spreading. This may facilitate the formation of local hubs of H3K9me and its associated factors. Consequently, the observed decrease in transcription could contribute to the instability of the repeats. An alternative explanation to the transcriptional changes observed at the boundaries could be that they are not the cause but rather the consequence. It is suggested that the presence of a demethylase at the boundaries of regions limits the extension of heterochromatin (Aygun et al, 2013; Ragunathan et al, 2015), such that its absence affects the levels of H3K9me3 in the repeats, thus facilitating their transcription and inactivation at the boundary regions. Thus, heterochromatin can itself recruit the inhibitors that

limit its own spreading. A critical factor for efficient spreading is an optimal density of H3K9me3 to support the read-write mechanism for heterochromatin spreading (Cutter DiPiazza et al, 2021; Ragunathan et al, 2015). Therefore, the presence of a demethylase that limits the effective local levels of H3K9me3 could potentially restrict its spreading, as previously suggested (Aygun et al, 2013; Ragunathan et al, 2015). This demethylation allows for the maintenance of high levels of H3K9me3 within the repeats. Consequently, correct levels of H3K9me3 help to balance the components necessary for the write/read system, which are required for the formation of the H3K9me3 hub responsible for heterochromatin stability. These mechanisms might act in paralell to ensure the PcH stability.

Interestingly, the PHF2 homologous JmjC domain-containing protein Epe1 has been shown to prevent heterochromatin instability in *Schizosaccharomyces pombe* (Trewick et al, 2007; Zofall and Grewal, 2006). Consistent with our findings, Epe1 is essential for heterochromatin integrity since its loss impaired silencing at centromeres. Moreover, Epe1 restricts heterochromatin domain (Audergon et al, 2015; Ayoub et al, 2003; Trewick et al, 2007; Zofall and Grewal, 2006), although its catalytic activity has not been confirmed. In mammals KDM4B antagonizes the H3K9 methylation, in this case trimethylation, at PcH heterochromatin (Decombe et al, 2021; Fodor et al, 2006).

Other alternative mechanisms may contribute to PHF2-mediated heterochromatin stability. It has been observed that, the positioning of heterochromatin domains at the nuclear periphery facilitates heterochromatin nucleation and spreading. It is believed that peripheral tethering creates a specialized nuclear subdomain that promotes the efficient loading of factors involved in histone turnover suppression. Loss of factors involved in peripheral tethering has been shown to increase histone turnover, leading to a reduction in H3K9me3 density and defective heterochromatin propagation (Holla et al, 2020; Murawska et al, 2021; Noma et al, 2001). On this basis, it would be interesting to test whether PHF2 modulates histone turnover promoting the maintenance of H3K9me3 density and heterochromatin stability. Notably, the major histone mark associated with lamina association is H3K9me2, which is targeted by PHF2. Therefore, the loss of PHF2 could disrupt the anchoring of heterochromatin to the lamina, resulting in heterochromatin destabilization. In addition, previous studies from our laboratory have demonstrated the interaction between PHF2 and E2F, a key proliferation factor in neural progenitors (Pappa et al, 2019). Interestingly, it has been proposed that E2F1 contributes to heterochromatin stability by recruiting enhancer of zeste homolog 2 (EZH2) to various repeat sequences, including simple repeats, satellites, and LINEs (Ishak et al, 2016).

## Accumulation of DNA damage in PHF2 depleted cells

Our findings demonstrate that depletion of PHF2 is associated with the accumulation of DNA DSBs, DNA damage, and genome instability. One major contributing factor to DNA damage accumulation is the unscheduled changes in transcription dynamics. In the case of PHF2 depletion, unscheduled transcription from pericentromeric heterochromatin repeats can lead to collisions between the replication and transcription machineries, resulting in DNA breaks and R-loops accumulation (Bayona-Feliu

et al, 2017; Hoffman et al, 2015; Merrikh et al, 2011). Furthermore, the known role of PHF2 in DNA repair, suggests that its depletion could contribute to the accumulation of DSBs and genome instability (Alonso-de Vega et al, 2020). In addition, it has been proposed that the demethylation of H3K9 may be an important step in the repair of DSBs. Thus, the global increase in H3K9me3 observed upon PHF2 depletion may impede DNA repair, as reported for the KDM4B demethylase (Colmenares et al, 2017; Young et al, 2013; Zheng et al, 2014).

Our research lays the foundation for further exploration into the role of H3K9 methylation in maintaining genomic stability and regulating gene expression in other cellular contexts. Specifically, the involvement of PHF2 in cancer has been extensively described. On the other hand, several studies have proposed targeting enzymes that modify H3K9me as potential therapeutic interventions in cancer treatment (Kondo et al, 2008; Wagner and Jung, 2012). However, our findings, combined with previous studies on the removal or alteration of H3K9me2/3 histone methyltransferases (HMTs) (Zeller et al, 2016), suggest that modifying or eliminating H3K9 methylation may not be an effective therapeutic strategy. Both scenarios have the potential to induce genomic instability, which poses a significant drawback in these treatment approaches.

In summary, our work deepens our understanding of chromatin dynamics and their implications for genome stability, development and diseases.

# Methods

## Cell culture

Mouse neural stem cells (NSCs) were obtained from cerebral cortices of C57BL/6J mouse fetal brains (E12.5). They were maintained in medium containing equal parts of DMEM F12 (without Phenol Red, Gibco) and Neurobasal medium (Gibco) with Glutamax (1%), N2, and B27 supplements (Gibco), Penicillin/Streptomycin, sodium pyruvate (1 mM), Heparin (2 mg/l), non-essential amino acids (0.1 mM), Hepes (5 mM), bovine serum albumin (BSA) (25 mg/l), and β-mercaptoethanol (0.01 mM) as detailed (Estaras et al, 2012; Iacobucci et al, 2021). Fibroblast Growth Factor (FGF) (Invitrogen) and recombinant human Epidermal Growth Factor (EGF) (R&D Systems) to 10 ng/ml and 20 ng/ml final concentrations, respectively, were included into the media. NSCs were cultured in Poly-D-lysine (5 μg/ml, 2 h 37 °C) and laminin (5 μg/ml 37 °C, 4 h 37 °C) precoated dishes as previously described (Currle et al, 2007). Growing in these conditions, they preserve the ability to self-renew and to generate a wide range of differentiated neural cell types under appropriated conditions (Currle et al, 2007; Iacobucci et al, 2021; Pollard et al, 2006). Human HEK 293T and mouse NIH3T3 cells were cultured under standard conditions (Blanco-Garcia et al, 2009) in Dulbecco's modified Eagle's medium supplemented with 10% fetal bovine serum. The cell lines were recently authenticated and frequently tested for mycoplasma contamination.

## Primary antibodies and reagents

Antibodies used were recognized: PHF2 (Cell Signaling, D45A2), PHF8 (Abcam, ab36068), H3K9me2 (Abcam, ab1220), H3K9me3

(Abcam, ab8898), KIAA1718 (Abcam, ab36044), H3K4me3 (Abcam, ab8580), gamma phospho-Histone H2A.X (Ser139) (Merck, 05-636), Lamin B1 (Proteintech_66095-1-lg), GAPDH (Synaptic Systems, 247002), β-TUBULIN (Millipore, #MAB3408), HP1 alpha (Abcam, ab77256), HA (Abcam, ab20084), FLAG (Sigma, F3165), unspecific IgGs (Diagenode, C15410206), Anti-Rabbit Alexa fluor 488 (ThermoFisher #A-11008), Anti-Rabbit Alexa fluor (ThermoFisher, #A-11011), Anti-Rabbit Alexa fluor 647 (ThermoFisher, #A-21245), Anti-Mouse Alexa fluor 555 (ThermoFisher, #A28180), and DAPI (ThermoFisher, D1306).

## Plasmids and recombinant proteins

Previously published specific lentiviral vectors were either purchased from Sigma or cloned in pLKO.1 puro vector using AgeI and EcoRI sites, brackets indicate target sequence: pLKO-random (CAACAA-GATGAAGAGCACC) and pLKO-mPHF2_1 (CGTGG CTATTAAAGTGTTCTA), pLKO-mPHF2_2 (CCTTATCCACCT CCCACT-TGACC). P3xFlag-PHF2 human cDNA was kindly provided by Drs Jiemin Wong, and Jiwen Li. PHF2 catalytic mutant (249HI251D > AIA) was previously generated (Pappa et al, 2019). p3xFlag-PHF2 plasmid was used to obtain PHF2ΔPHD mutant by using the restriction enzymes HindIII and PasI. Finally, PHF2ΔCharged was described in (Vicioso-Mantis et al, 2022). pLIV-PHF2, and pLIV-PHF2ΔCharged plasmids were constructed obtaining the coding sequence using BamHI restriction enzyme from previous p3xFlag plasmids, and were introduced into pLIV plasmid open with the same restriction enzyme. pLIV- PHF2ΔPHD plasmid was constructed following the same strategy as per p3xFlag-PHF2ΔPHD. pInducer-PHF2 HD/AA was described in (Pappa et al, 2019). p3xFlag-MaSat-NLS-ZF18-mVenusN and p3xFlag-mVenusC-HP1b-chromodomain plasmids were kindly provided by Drs A Jeltsch and I Amelio (Lungu et al, 2017; Panatta et al, 2022). All mutants were checked by electrophoresis and sequencing.

## Lentiviral transduction

Lentiviral transduction was carried out as previously described (Asensio-Juan et al, 2017). Basically, HEK 293T cells were transfected with a mix of packaging, envelop, and shRNA transfer vector DNAs or other expression vectors (1, 3, and 4 μg, respectively). 24–30 h later, the medium was collected and the virus were used to transduce NIH3T3 cells. For NSCs, viruses were concentrated by ultracentrifugation (26,000 rpm, 2.5 h at 4 °C). Viral particles were then used to transduce NSCs. After 24 h, cells were selected with puromycin (2 μg/ml) (Merck, P8833). After selection between 99–100% the cells express the shRNA.

## Chromatin immunoprecipitation (ChIP) assays

ChIP assays were performed as described (Valls et al, 2007) with adaptations: $1 \times 10^6$ NSCs were cross-linked with Cross-link Gold 0.4% (Diagenode, C01019027) in PBS 30 min, and then fixed with methanol-free formaldehyde 1% for 10 min and stopped with 0.125 mM glycine for 10 min. Cells were lysed in lysis buffer 1 (50 mM Hepes; 140 mM NaCl; 1 mM EDTA; 10% Glycerol; 0.5% NP-40; 0.25% TX-100), lysis buffer 2 (10 mM Tris; 200 mM NaCl; 1 mM EDTA; 0.5 mM EGTA) and lysis buffer 3 (10 mM Tris, 100 mM NaCl; 1 mM EDTA; 0.5 mM EGTA; 0.1% Na-

Deoxycholate; 0.5% N-Lauroylsarcosine). A bioruptor sonicator was used for chromatin fragmentation before immunoprecipitation. The immuno-complex was captured using magnetic beads (Magna ChIP™ Protein A Magnetic Beads Millipore 16-661). After DNA purification by phenol-chloroform extraction, ChIP DNA was quantified by qPCR with SYBR Green (Roche) in a QuantStudio5 (Applied Biosystems) using primers indicated in Appendix Table S3.

## Indirect immunofluorescence, image acquisition, and quantification

Immunofluorescence assays were accomplished as described (Sanchez-Molina et al, 2014). Basically, cells were fixed for 20 min in 4% paraformaldehyde and washed with PBS-Triton X-100 (0.1%). Methanol treatment was done for 10 min before blocking at room temperature for 1 h in 1% BSA (in PBS with 0.1% Triton X-100). Incubation with the primary antibodies in blocking solution was performed overnight at 4 °C. The following day, cells were incubated for 2 h at room temperature with Alexa-conjugated secondary antibodies, and after washes, cells were incubated with DAPI 0.1 (ηg/μl) (Sigma) for 1 h, then mounted with ProLong Glass Antifade Mountant (Thermofisher, P36980). Confocal images were captured by Leica SP5 confocal microscope using LAS-AF software. Super-resolution images were acquired in a Dragonfly 505 multimodal spinning-disk confocal microscope (Andor Technologies, Inc). For live cell imaging cells were cultured in an IBIDI ® μ-35 mm dish, co-transfected with the VenusN and VenusC reporter vectors that bind MajorSat and H3K9me3, respectively, and imaged using the specific stage by Andor. Laser excitation and image acquisition were performed sequentially for each channel. Exposure duration and laser potency were adjusted accordingly, ensuring the absence of oversaturated pixels. Confocal 3D images were captured as Z-stacks with intervals of 0.13 μm. Super-Resolution Radial Fluctuations (SRRF) algorithm (Culley et al, 2018; Gustafsson et al, 2016) was applied using the SRRF-Stream+ module (Andor) operated from the Fusion software. The following parameters were used for all conditions: 1x ring radius, 5x radiality magnification, 100 frames. Confocal images are showed as Z-projections of maximum intensity. Fluorescence intensity per cell was measured using FIJI ImageJ Software. The cell fluorescence was corrected with the formula *Integrated Density − (Area of selected cell × Mean fluorescence of background readings)*. For nuclear foci quantification, the *3D Objects Counter* tool was used with an average threshold of 50, then number and volume measurements were taken. For nuclear distance measurements, the *Radial profile* plugin was used, and fluorescence profile was plotted. Then, linear measure in microns from a clear fluorescence peak between channels was quantified.

## RNA extraction and qPCR

RNA extraction was achieved using TRIzol reagent (Thermofisher, 15596018) following the manufacturer instructions. Reverse transcription was performed with 2 μg of RNA using High Capacity cDNA reverse transcription kit (Thermofisher, 4368814). qPCRs were performed using SYBR Green (Roche) in QuantStudio5 (Applied Biosystems). Primer pairs used are included in Appendix Table S3.

## Protein extraction and western blot

For total protein extraction RIPA buffer supplemented with protease inhibitors was used. Immunoblotting was performed using standard procedures in SDS-PAGE (10% for PHF2 and 15% for histones), transferred to a Nitrocellulose membrane and incubated with primary antibody overnight at 4 °C. Then, secondary antibodies conjugated with fluorescence (IRDye 680RD goat anti-rabbit IgG, Li-Cor 926-68071; IRDye 800CW goat anti-mouse IgG, Li-Cor 926-32210), and visualized in the double channel IR scanner Li-Cor Odissey machine model ODY-1871. ImageJ software was used for immunoblot relative quantification following NIH guidelines.

## Co-immunoprecipitation procedure (CoIP)

For this experiment, we overexpressed both PHF2 and either HP1BP3 proteins in HEK293T cells. For protein extraction, IPH buffer (50 mM Tris-HCl pH 7.5, 150 mM NaCl, 5 mM EDTA, 0.5% NP-40) was used. Then, a specific antibody is added to immunoprecipitate the target protein, as well as immunoprecipitation with IgGs of the same species in order to control the unspecific binding of the sample. After overnight incubation at 4 °C, agarose beads of protein A (Millipore 16-125) or protein G (Millipore IP04) are used and incubated for 4 h at 4 °C in a rotating wheel. Following that, the beads are recovered and eluted by shaking at 900 rpm and 55 °C for 15 min. Finally, immunoprecipitated proteins are denatured to proceed with electrophoresis and Western Blot.

## Mass spectrometry

Total protein extract was performed as described for co-immunoprecipitation, and anti-PHF2 specific antibody was used to immunoprecipitate the endogenous PHF2 protein in control and PHF2-deleted NSCs, followed by agarose beads incubation. IgGs were also used as a control. Samples were reduced, alkylated, and diluted for trypsin digestion (1 μg, 37 °C, 8 h, Promega, V5113). After digestion, peptide mix was acidified and desalted with a MicroSpin C18 column (The Nest Group, Inc). Samples were analyzed using a LTQ-Oribtrap Velos Pro mass spectrometer (ThermoFisher, San Jose, CA, USA), coupled to an EasyLC 1000 (ThermoFisher Proxeon, Odense, Denmark). The mass spectrometer was operated in positive ionization mode with nanospray voltage set at 2.1 kV and source temperature at 300 °C. The acquisition was performed in data-dependent acquisition (DDA) mode and full MS scans with 1 micro scans at resolution of 60,000 were used over a mass range of $m/z$ 350–2000 with detection in the Orbitrap. All data were acquired with Xcalibur software v2.2, and analyzed using the Proteome Discoverer software suite (v1.4, ThermoFisher) and the Mascot search engine (v2.5, Matrix Science). Data was searched against Swiss-Prot mouse database and false discovery rate (FDR) was set to a maximum of 5%.

## ChIP-seq procedure

ChIPs were performed as described (Fueyo et al, 2018). Basically, sonication was done using a Bioruptor sonicator. 10% of the total material was reserved as the input sample. Primary antibodies (Abcam, ab8898; Cell Signaling D45A2, Abcam ab8580) were used

for immunoprecipitation. Decrosslinked DNA was purified by phenol-chloroform precipitation. The libraries were prepared and amplified to make DNA NanoBall (DNB) and were loaded into a nanoarray and single-end 50 base pairs reads were generated by combinatorial Probe-Anchor synthesis (cPAS). Single-end reads were aligned to the mouse mm10 reference genome using Bowtie2 (v2.3.5.1) (Langmead and Salzberg, 2012) with default options. SAMtools (v1.9) (Li et al, 2009) was used to filter out unmapped reads, duplicates, and low-quality alignments using the flag 3844. The resulting BAM files were sorted and deepTools (Ramirez et al, 2016) was used to generate input-subtracted and counts per million (CPM) normalized signal tracks (bamCompare --operation subtract --normalizeUsing CPM --scaleFactorsMethod None) in bedgGraph and bigWig format.

## Assay for transposase accessible chromatin (ATAC-seq) procedure

ATAC-seq was performed as previously described (Buenrostro et al, 2013) with some modifications. Basically, $5 \times 10^6$ cells were scrapped and resuspended in RBS buffer (10 mM Tris-HCl pH 7.4; 10 mM NaCl; 3 mM MgCl$_2$) and then Igepal CA-630 0.1% was added to achieve cell lysis. Then 50,000 nuclei underwent Transposition reaction with Transposes enzyme and buffer kits (Illumina, FC-121-1030) and immediately following transposition, DNA was purified using Qiagen MinElute PCR Purification Kit (Qiagen, 28004). For library construction, Nextera DNA Library Prep Kit (FC-121-1030), along with unique dual index barcodes for each sample and NEBNext High-Fidelity 2x PCR Master Mix (New England Lab, M0541) were used. To reduce GC and size bias in PCR, the PCR is monitored using a side qPCR to stop amplification prior to saturation. Finally, Ampure XP beads (Beckman Coulter, A63880) were prepared for size selection. Libraries were then pair-end sequenced using NextSeq2000 and approximately 50 M reads per sample were obtained. Data quality control and processing were done following nf-core and ENCODE ATAC-seq pipeline. Paired-end reads were aligned to the mouse mm10 reference genome using Bowtie2 (v2.3.5.1) (Langmead and Salzberg, 2012) and specifying the "--very-sensitive-local" mode.SAMtools (v1.9) (Li et al, 2009) was used to filter out the low-quality reads with the flag 1796, remove reads mapped in the mitochondrial chromosome, and discard those with a MAPQ score below 20. The resulting BAM files were sorted and deepTools (Ramirez et al, 2016) was used to generate counts per million (CPM) normalized signal tracks (bamCoverage-samFlagInclude 64-normalizeUsing CPM) in bedGraph and bigWig format.

## Downstream analysis

BEDTools (v2.28.0) (Quinlan and Hall, 2010) was used to calculate the mean ATAC-seq and ChIP-seq signal within TEtranscripts (Jin et al, 2015) mm10 repeat annotation or genome-wide 10 Kb bins (map -o mean). With regard to TEtranscripts annotation, it consists of a GTF file customly generated from the RepeatMasker annotation comprising more than 3.7 million repeats and it is organized into three levels: Class, Family, and Repeat (group of sequences sharing a specific structure). Repeats that overlap with regions included in the mm10 ENCODE BlackList were discarded from the analysis. Heatmaps were performed by using the R

package pheatmap. The "euclidean" distance measure and the "complete" cluster method were used in clustering rows and columns.

## Statistical analysis

Quantitative data were expressed as standard deviation (SD). The significance of differences between two groups was assessed using the Student's *t*-test. For non-parametric data, Wilcoxon signed-rank test or Mann–Whitney U test were used, according to the needs of the analyses. Fisher's exact test test was assessed for evaluating the difference between mono and multinucleated cells. ANOVA or Kruskal–Wallis test were applied for three or more data sets ($*p < 0.05$, $**p < 0.01$, $***p < 0.001$, $****p < 0.0001$).

The experiments were conducted blindly.

## Data availability

All sequenced data have been deposited in the GEO database under the super series GSE242385. ChIP-seq data have been deposited in the GEO database under the accession GSE242383. ATAC-seq data have been deposited in the GEO database under the accession GSE242384. Mass spectrometry data have been deposited to the ProteomeXchange consortium via the PRIDE (Perez-Riverol et al, 2022) partner repository under the dataset identifier PXD051336. Previously deposited RNA-seq, H3K9me2-seq, H1-seq, and RNAPolII-seq are available at GSE122264, GSE122263, GSE49564, and GSE66961, respectively.The source data of this paper are collected in the following database record: biostudies:S-SCDT-10_1038-S44319-024-00178-7.

## Peer review information

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

## Acknowledgements

We would like to thank Drs Jiemin Wong, and Jiwen Li for the PHF2 plasmid. Dr T Jenuwein for pCMV-FLAG-SUV39H1 plasmid. Dr B Garfinkel for pCDNA3.1-HPIBP3-HA plasmid. Drs I Amelio and A Jeltsch for p3xFlag-MaSat-NLS-ZF18-mVenusN and p3xFlag-mVenusC-HP1b-chromodomain plasmids. Drs J. Bernues, F. Azorín, A. Vaquero, C. Gallego, M. Arbonés, M. Beato, and S Sanchez-Molina for reagents and suggestions. All member of the team for helpful comments. This study was supported by grants BFU2015-69248-P, PGC2018-096082-B-I00 and PID2021-125862NB-I00 to MAMB from the Spanish Ministry of Economy to MAMB, FPI grant PRE2019-087498 to SAI.

## Author contributions

**Samuel Aguirre**: Formal analysis; Validation; Investigation; Writing—review and editing. **Stella Pappa**: Conceptualization; Formal analysis; Investigation; Writing—review and editing. **Núria Serna-Pujol**: Data curation; Software; Formal analysis; Investigation; Writing—review and editing. **Natalia Padilla**: Data curation; Software; Formal analysis; Writing—review and editing. **Simona Iacobucci**: Investigation; Writing—review and editing. **A Silvina Nacht**: Investigation; Writing—review and editing. **Guillermo P Vicent**: Investigation; Writing—review and editing. **Albert Jordan**: Conceptualization; Writing—review and editing. **Xavier de la Cruz**: Conceptualization; Software; Formal analysis. **Marian A Martínez-Balbas**: Conceptualization; Resources; Formal analysis; Supervision; Funding acquisition; Methodology; Writing—original draft; Project administration.

Source data underlying figure panels in this paper may have individual authorship assigned. Where available, figure panel/source data authorship is listed in the following database record: biostudies:S-SCDT-10_1038-S44319-024-00178-7.

## Disclosure and competing interests statement

The authors declare no competing interests.

# Expanded View Figures

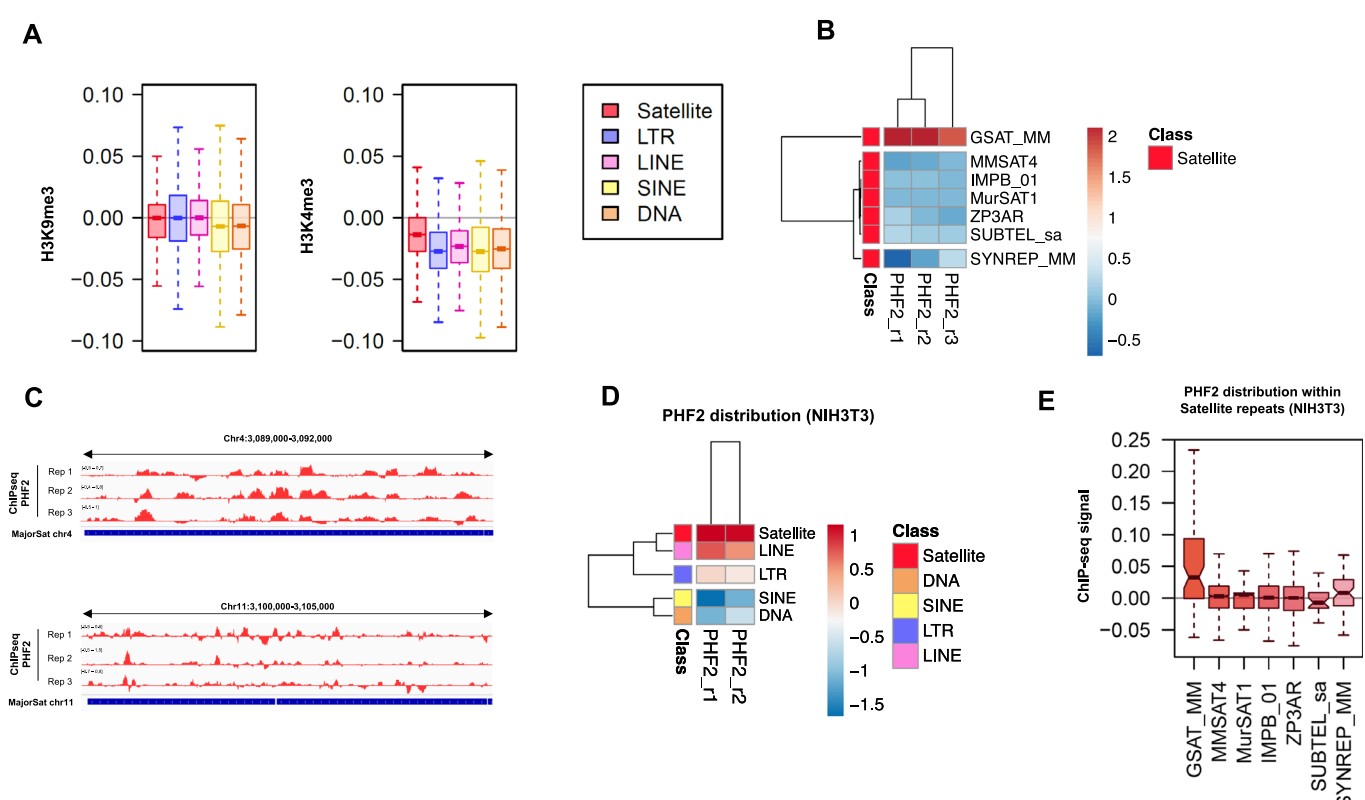

**Figure EV1. PHF2 is enriched in satellites repeats.**

(A) Boxplot showing the H3K9me3 and H3K4me3 input-subtracted ChIP-seq signal within Satellite, LTR, LINE, SINE, and DNA classes of repeats in control NSCs. The boxplots represent the distribution of the different samples' CPM-normalized and input-subtracted ChIP-seq signal (y-axis) in the corresponding group of repeats (x-axis). Box plots: centerlines show the medians; box limits indicate the 25th and 75th percentiles; whiskers extend to the minimum and maximum. Data used for this analysis correspond to the averaged values obtained from two independent ChIP-seq replicates. Exact sample sizes for each group: Satellite = 36234; LTR = 970039; LINE = 987285; SINE = 1527608; DNA = 162787. (B) Heatmap and clustering of the mean input-subtracted ChIP-seq signal of three independent PHF2 replicates within seven groups of repeats belonging to the Satellite class. Box plots: centerlines show the medians; box limits indicate the 25th and 75th percentiles; whiskers extend to the minimum and maximum. Exact sample sizes for each group: GSAT_MM = 77; MMSAT4 = 1597; MurSAT1 = 5398; IMPB_01 = 26485; ZP3AR = 2570; SUBTEL_sa = 31; SYNREP_MM = 69. (C) IGV genome browser screenshots illustrating the continuous input-subtracted quantification of three replicates of PHF2 ChIP-seq samples within the major satellite of chromosmes 4 and 11. (D) Heatmap and clustering of the mean input-subtracted ChIP-seq signal of two independent PHF2 replicates in NIH3T3 cells in Satellite, DNA, SINE, LTR, and LINE classes of repeats. (E) Boxplot showing the PHF2 input-subtracted ChIP-seq signal in NIH3T3 cells within seven groups of repeats belonging to Satellite class in shCT condition. The boxplots represent the distribution of the different samples' CPM-normalized and input-subtracted ChIP-seq signal (y-axis) in the corresponding group of repeats. Box plots: centerlines show the medians; box limits indicate the 25th and 75th percentiles; whiskers extend to the minimum and maximum. Data used for this analysis correspond to the averaged values obtained from two independent PHF2 ChIP-seq replicates in NIH3T3 cells. Exact sample sizes for each group: GSAT_MM = 77; MMSAT4 = 1597; MurSAT1 = 5398; IMPB_01 = 26485; ZP3AR = 2570; SUBTEL_sa = 31; SYNREP_MM = 69.

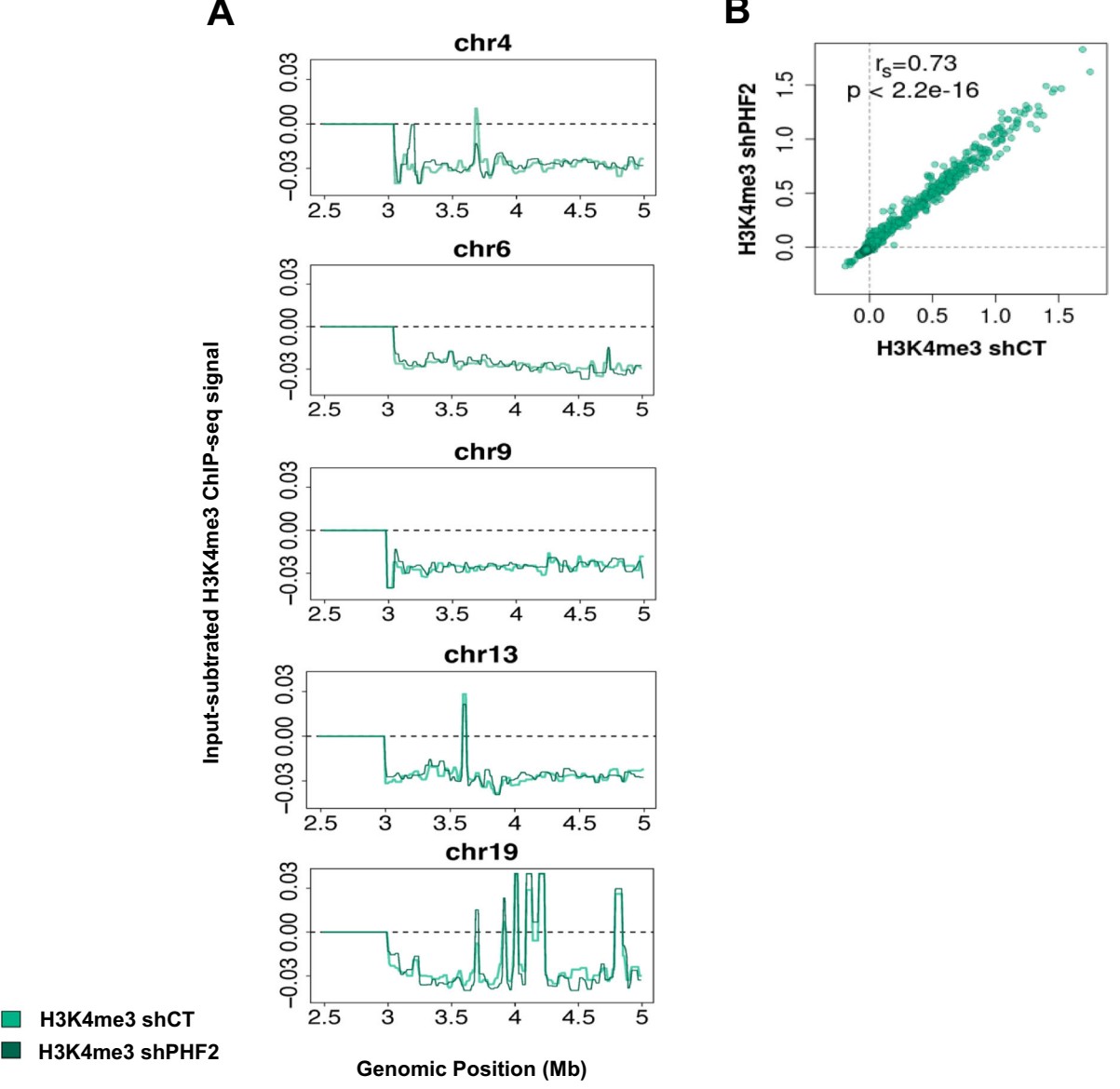

**Figure EV2.  PHF2 balances H3K9me3 at Pc boundaries.**

(A) Line plot showing the H3K4me3 profile across the first 5 Mb (2.5–5 Mb) from the centromeric end of chromosomes 4, 6, 9, 13, and 19 in control and PHF2-depleted NSCs. Data used for this analysis correspond to the averaged input-subtracted signal obtained from two independent H3K4me3 ChIP-seq replicates. (B) Spearman's correlation between H3K4me3 input-substracted ChIP-seq signal in control and PHF2-depleted NSCs. Data used for this analysis correspond to the averaged input-subtracted values within 10 Kb bins along the whole genome obtained from two independent H3K4me3 ChIP-seq replicates.

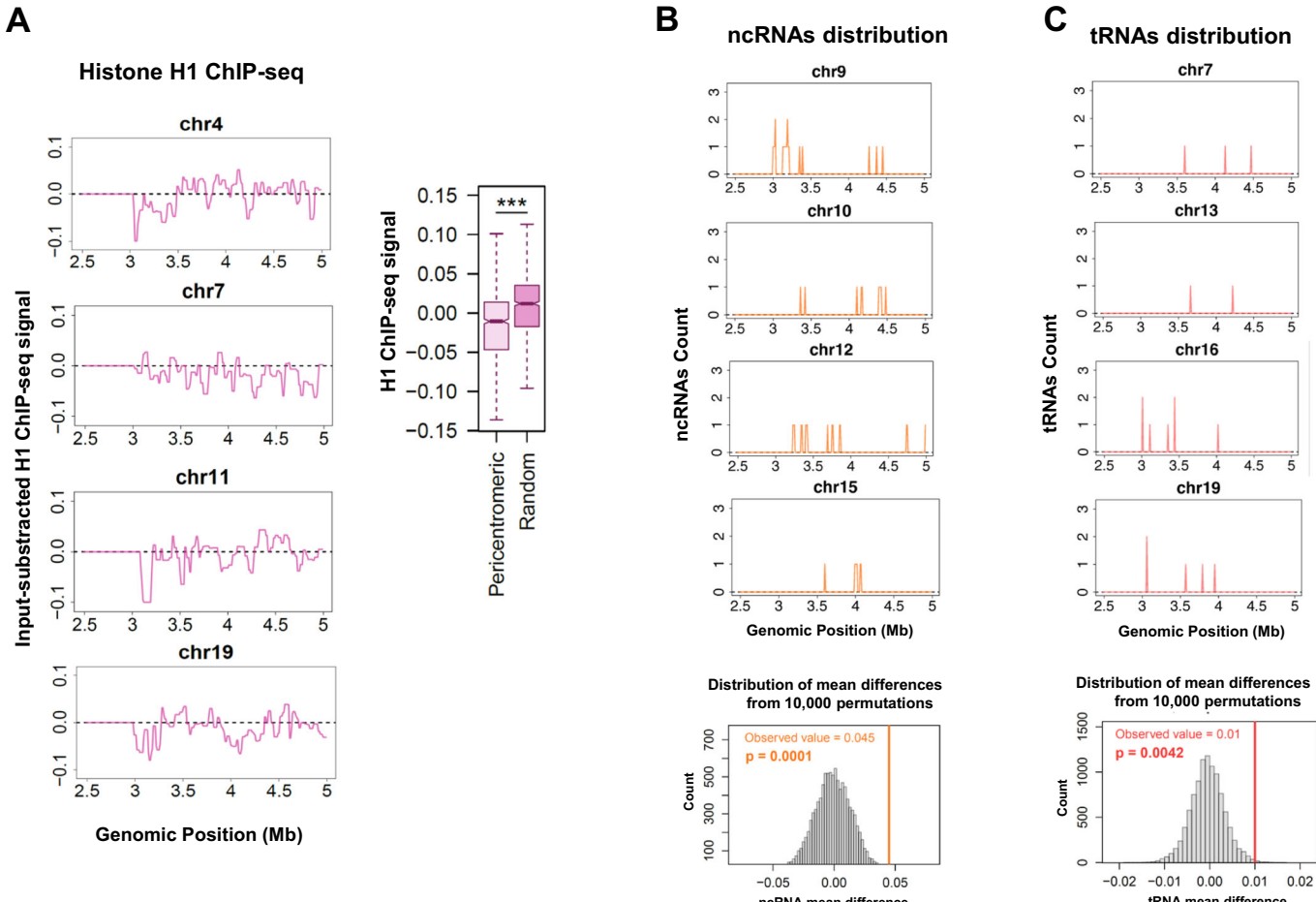

**Figure EV3. PcH boundaries characterization.**

(A) Line plot showing the histone H1 (GSM1199586) profile across the first 5 Mb (2.5–5 Mb) from the centromeric end of chromosomes 4, 7, 11, and 19 in mESCs. Data used for this analysis correspond to the averaged input-subtracted ChIP-seq signal (left panel). Boxplot displaying the histone H1 input-subtracted ChIP-seq signal within 1000 pericentromeric bins (3–3.5 Mb) from all chromosomes compared to 1000 random bins. The boxplots represent the distribution of the sample's CPM-normalized and input-subtracted ChIP-seq signal (y-axis) in the corresponding group of bins (x-axis). Box plots: centerlines show the medians; box limits indicate the 25th and 75th percentiles; whiskers extend to the minimum and maximum. ***$p < 0.001$ (Mann– Whitney U test). Exact sample sizes for each group: Pericentromeric $= 1000$; Random $= 1000$ (right panel). (B) Line plot showing the count distribution of ncRNAs (Ensembl GRCm38.86 annotation) across the first 5 Mb (2.5–5 Mb) from the centromeric end of chromosomes 9, 10, 12, and 15 (upper panel). Histogram of the mean differences from the permutation test between ncRNAs count distribution in pericentromeric and random bins. A total of $N = 10,000$ permutations were performed. The orange vertical bar corresponds to the observed difference in medians between ncRNAs count distribution in pericentromeric and random bins (lower panel). (C) Line plot showing the count distribution of tRNAs (Genomic tRNA Database, GtRNAdb, annotation) across the first 5 Mb (2.5–5 Mb) from the centromeric end of chromosomes 7, 13, 16, and 19 (upper panel). Histogram of the mean differences from the permutation test between tRNAs count distribution in pericentromeric and random bins. A total of $N = 10,000$ permutations were performed. The pink vertical bar corresponds to the observed difference in medians between tRNAs count distribution in pericentromeric and random bins (lower panel).

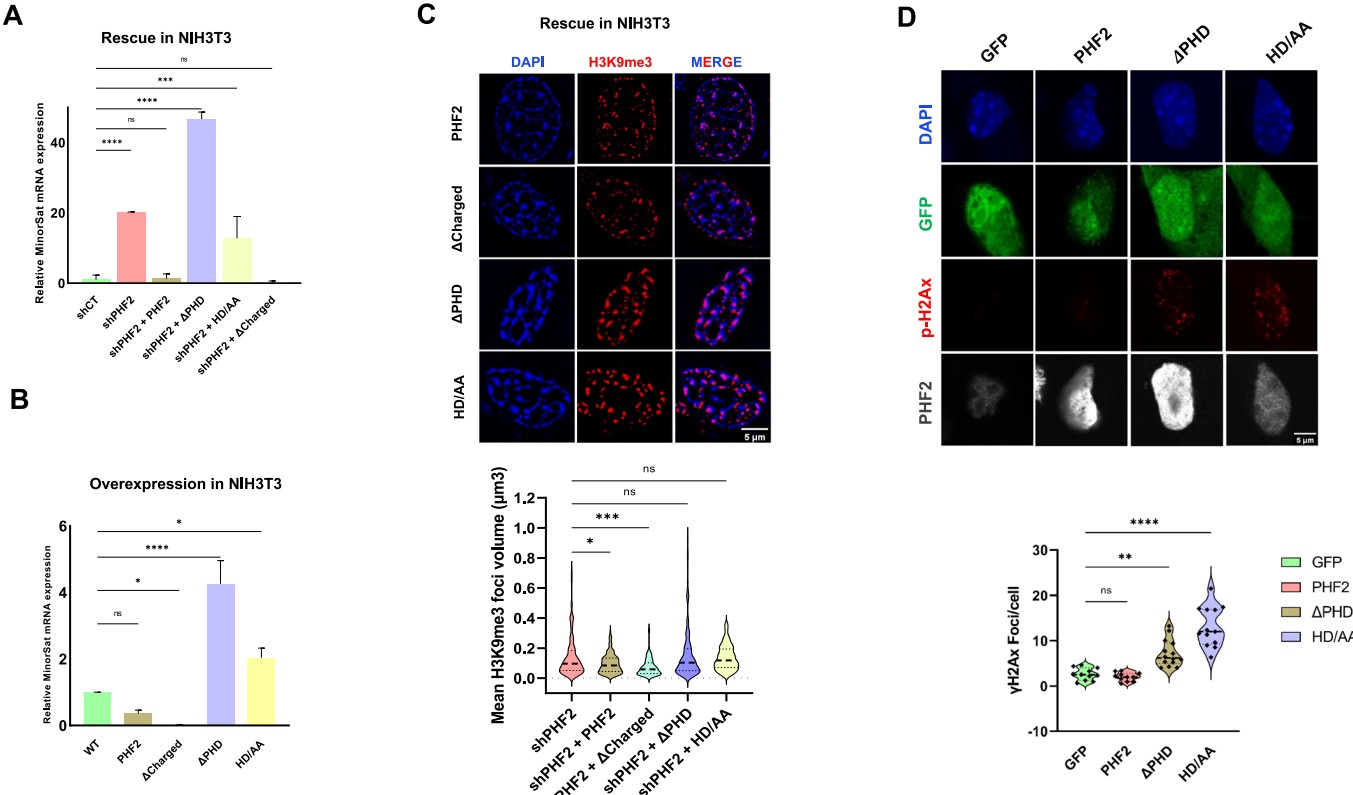

**Figure EV4. The maintenance of PcH stability relies on the PHD and JmjC domains within PHF2.**

(A) PHF2-depleted NIH3T3 were transfected with PHF2 WT, ΔCharged, ΔPHD, or HD/AA mutants. Total RNA was prepared and the levels of RNA levels of major satellite, minor satellite, were determined by qPCR. Mean was calculated from data of 3 biologically independent experiments and normalized to *Gapdh* reference gene levels and figure shows values relative to shCT cells. Error bars indicate SD. ***$p < 0.001$, ****$p < 0.0001$ (Student's t-test). (B) NIH3T3 cells were transfected with PHF2 WT, ΔCharged, ΔPHD, or HD/AA mutants. Total RNA was prepared and the levels of RNA levels of major minor satellite, were determined by qPCR. Mean was calculated from data of 3 biologically independent experiments and normalized to *Gapdh* reference gene levels and figure shows values relative to shCT cells. Error bars indicate SD. *$p < 0.05$, ***$p < 0.001$ (Student's t-test). (C) PHF2-depleted NIH3T3 cells were transfected with PHF2 WT, ΔCharged, ΔPHD, or HD/AA mutants. Super resolution (SRRF) immunostaining is depicted using anti H3K9me3 antibody and DAPI. The volume of the H3K9me3 foci were determined on confocal images using ImageJ software (see Methods). Data shown are representative of three biologically independent experiments. Scale bar indicates 5 μm. Violin plots represent the foci volume quantification of H3K9me3 foci of 100 cells ($n = 100$). *$p < 0.05$; ***$p < 0.001$ (Student's t-test). (D) NSCs were nucleofected either with GFP alone or with PHF2 WT, ΔPHD, or HD/AA mutants. Cells were fixed and stained with PHF2 and γH2Ax antibodies and DAPI. Green cells were analyzed. More than 30 cells were quantified. Data shown are representative of 3 biologically independent experiments. Scale bar indicates 5 μm. Violin plots represent the number of γH2Ax foci/cell. **$p < 0.01$, ****$p < 0.0001$ (Student's t-test).

