## [Peer Review File · EMBO Reports]

PHF2-mediated H3K9me balance orchestrates heterochromatin stability and neural progenitor proliferation

Samuel Aguirre, Stella Pappa, Núria Serna-Pujol, Natalia Padilla, Simona Iacobucci, A. Silvina Nacht, Guillermo Vicent, Albert Jordan, Xavier de la Cruz, and Marian Martínez-Balbas

Corresponding author(s): Marian Martínez-Balbas (mmbbmc@ibmb.csic.es)

Review Timeline:

Submission Date:	30th Nov 23
Editorial Decision:	12th Jan 24
Revision Received:	13th Apr 24
Editorial Decision:	13th May 24
Revision Received:	18th May 24
Accepted:	4th Jun 24

Editor: Esther Schnapp

Transaction Report:

Dear Dr. Martínez-Balbas,

Thank you for the submission of your manuscript to EMBO reports. We have now received the full set of referee reports that is pasted below.

As you will see, the referees acknowledge that the findings are potentially interesting. However, they also point out that the data are not strong enough to support the main conclusions, and suggest several experiments to strengthen the study. Given the several points raised, it might be best to discuss the exact revision requirements in a video chat, if you like. You can of course also send me a proposed revision plan that we could discuss further, if necessary. Please let me know how you would like to proceed.

If we can agree on a revisions plan that addresses the referee concerns, I would like to invite you to revise your manuscript with the understanding that the referee concerns must be fully addressed and their suggestions taken on board. Please address all referee concerns in a complete point-by-point response. Acceptance of the manuscript will depend on a positive outcome of a second round of review. It is EMBO reports policy to allow a single round of major revision only and acceptance or rejection of the manuscript will therefore depend on the completeness of your responses included in the next, final version of the manuscript.

We realize that it is difficult to revise to a specific deadline. In the interest of protecting the conceptual advance provided by the work, we recommend a revision within 3 months (13th Apr 2024). Please discuss the revision progress ahead of this time with the editor if you require more time to complete the revisions.

- 1) A data availability section providing access to data deposited in public databases is missing. If you have not deposited any data, please add a sentence to the data availability section that explains that.
- 2) Your manuscript contains statistics and error bars based on $n=2$. Please use scatter blots in these cases. No statistics should be calculated if $n=2$.

3) We replaced Supplementary Information with Expanded View (EV) Figures and Tables that are collapsible/expandable online. A maximum of 5 EV Figures can be typeset. EV Figures should be cited as 'Figure EV1, Figure EV2' etc... in the text and their respective legends should be included in the main text after the legends of regular figures.

5) a complete author checklist, which you can download from our author guidelines <https://www.embopress.org/page/journal/14693178/authorguide>. Please insert information in the checklist that is also reflected in the manuscript. The completed author checklist will also be part of the RPF.

6) Please note that all corresponding authors are required to supply an ORCID ID for their name upon submission of a revised manuscript (<https://orcid.org/>). Please find instructions on how to link your ORCID ID to your account in our manuscript tracking system in our Author guidelines

<<https://www.embopress.org/page/journal/14693178/authorguide#authorshippinguidelines>>

I look forward to seeing a revised form of your manuscript when it is ready.

Referee #1:

Within the manuscript "PHF2-mediated H3K9me balance orchestrates heterochromatin stability and progenitor proliferation in early neurogenesis", by Aguirre, Pappa et al, the authors investigate the role of the histone demethylase PHF2 in heterochromatin stability. Mainly using Neural Stem Cells (NSC) as a model, the authors use ChIP-seq to suggest that PHF2 is located at the boundaries of pericentromeric heterochromatin (PcH). By perturbing the levels of PHF2, they provide evidence that PHF2, including at least its catalytic and PHD domains, functions at these boundaries to avoid spreading of the heterochromatin, thereby maintaining high levels of H3K9me3 and silencing of PcH.

The manuscript is well written, and the figures are illustrative. However, I have concerns about the robustness of the data analyses underlying the conclusions. The authors mainly use ChIP-Seq/ ATAC-seq and Immunofluorescence (IF). With regards to ChIP-seq: While ChIP-seq is robust to detect protein binding at individual loci, calling quantitative differences (and in particular over larger genomic bins like the authors apply here) and ChIP-seq repeat-analyses are much challenging and error-prone, among others due to (1) differences in signal-to-noise between individual ChIP-seq experiments; and (2) potential differences in abundance of the ChIPped protein. The first could be addressed by performing sufficient replicates (including ChIP-qPCR), the latter using spike-in ChIP-seq. Neither have been applied by the authors (while both should), and as such I do find the ChIP-seq analysis far from convincing. As a side note, it would be helpful if additional data is provided showing the quality and success of the individual ChIP-seq experiments, in particular by showing genome browser views showing the quality of the data, and relevant non-binned changes at the loci under investigation. The above are even more important as the ChIP-qPCR presented by the authors show extremely low recovery percentages (below 0.1% for positive loci, Fig 1e/ S1b). This is important as the other main read-out of the authors represent IF. And while the authors generally provide sufficient statistics for their IF experiments, also IF is prone to high variability (e.g. variation between replicates and cell cultures; and variation within slides). Also, the authors should validate their main findings in another cell line (potentially NIH3T3 which is already used in few experiments).

With further regards to data analysis: also data obtained from the PHF2 co-IP-MS experiments are poorly analyzed and presented (Fig 1b). While this pulldown should yield a list of interactors, often represented in a Volcano plot, only one interactor (HPIBP3) is now shown, together with some unclear associated scores. Also, the known PHF2 interactor SUV39H1/2 is apparently not detected by MS, but SUV39H1/2 is by a specific co-IP-Western? The authors should present the complete experiment with all PHF2 interactors, and interpret these data in the context of their paper.

Other main issues

- The authors mainly rely on NSCs (sometimes in combination with NIH3T3). But for the rescue experiments in Fig 6b, the authors only use NIH3T3 for unclear reasons. It would be helpful to perform this assay on NSCs as well, as these seem to behave slightly different than NSCs with regards to PHF2.
- A better characterization of the PHF2-associated boundaries would be helpful. What exactly happens if PHF2 is depleted from these sites? Also, I feel it is important to show what happens to gene expression at these boundaries, data which the authors refer to but excluded ("data not shown"). If there is no change in expression of genes at these boundaries, a clearer explanation of why not (as one would expect so if the heterochromatin starts spreading) is helpful.
- There is no spread of samples on the PC1 in Fig S4b; As such, this figure as it is now does not make sense.
- For the ChIP-seq repeat analysis Figs 1d, S1c, 4b and 4e, it is unclear which exact values the y-axis represent.
- The proteome data (Fig 2b) does not seem to be submitted to a public database.

Referee #2:

In this study, Samuel and collaborators investigated the role of PHF2 demethylase on heterochromatin integrity. Using knockdown / rescue approaches in the neural stem cells as a model they assessed the proteins that interact with PHF2, the impact on H3K9me3 distribution at pericentromeric regions (PcH), and how heterochromatin accessibility, transcription and integrity is affected. This study corresponds to a high amount of work and combines various techniques (IF, Co-IP Mass-spectrometry, ChIP-Seq/qPCR, ATAC-Seq) and demonstrates a new role of PHF2 at PcH in mammalian cells. However, this manuscript should go under modifications/improvement to meet the high-quality standard at EMBO Journal.

Comments and suggestions for modifications:

I do not understand the point of talking about genome stability before talking about repeated sequences that can recombine (Line 5 of the introduction). Could you change the position of this sentence in the introduction? Or elaborate more.

The neural stem cell (NSC) model and the focus on PHF2 protein are introduced a little too abruptly for me... Why NSC and not ESC? Why PHF2 and not a histone methyltransferase like G9a or Setdb1 (which are not mentioned as actors of the methylation of H3K9) or another histone demethylase? Please explain a little more or differently.

I am not convinced... I need more information about the relevance of the cellular model and the focus on this enzyme. Also, I am not in favor of the use of NIH3T3 cells line (a cell line with an abnormal karyotype and genome instability) to study heterochromatin stability...

I guess this study is a follow-up of the previous study of the authors (Pappa et al 2019) but readers still need more explanation to understand the purpose of this study and why it is relevant to study H3K9me2/me3 and PcH transcription/organization in the context of a histone H3K4 demethylase.

The authors should present more openly the data of ChIP analysis for H3K9me2 it's confusing in the text because they only talk clearly about H3K9me3 and H3K4me3? By the way, were the H3K9me3 and H3K4me3 data already published? If yes it should be explicitly written.

Page 8 last sentence: A reference is needed about the definition of chromocenters in mouse

In Fig S6B It would be more understandable to add representative images of the labeling (H3K9me3) of chromocenters like in Fig2. More explanation would be needed on the various degrees of alteration in the chromocenters foci.

Referee #3:

In the paper entitled "PHF2-mediated H3K9me balance orchestrates heterochromatin stability and progenitor proliferation in early neurogenesis" by Aguirre, Pappa et al, the authors study the H3K9 demethylase PHF2, extending previous work from the same lab (Pappa et al, 2019). They show that PHF2 interacts with the heterochromatin proteins HP1BP3 and SUV39, and that PHF2 loss leads to increased levels of transcripts from satellite repeats. They show that the PHD domain and the catalytic domain of PHF2 play a role in driving this increase, while the charged domain of PHF2 does not. The authors also study the localization of PHF2 using ChIP-seq and microscopy, extending the previous work in Pappa et al 2019 that showed that PHF2 binds transcription start sites. They conclude that PHF2 is enriched at pericentromeric boundaries, and propose that it acts as an anti-spreading factor. As described below, I am not convinced by this part of the work, as I do not see clear evidence in the data that PHF2 has a strong preference for pericentromeric boundaries and that it regulates heterochromatin spreading. Finally, the authors extend their previous work and analyse the link between PHF2 and DNA damage, putting a focus on satellite transcripts.

I think this is a potentially interesting paper, but the data about the localization of PHF2 seem to be somewhat contradictory, and the mechanistic conclusions are in my opinion highly speculative at this stage. In particular, regarding the localization of PHF2, Fig. 1B puts forward the idea that PHF2 interacts with heterochromatin proteins, and the ChIP analysis in Fig. 1D shows that PHF2 binds at least to some extent to heterochromatic satellite repeats. However, the microscopy images in Fig. 2C show that PHF2 does not localize at chromocenters. Localization at pericentromeric boundaries is not obvious from the microscopy images in Fig. 4C, and the selected line profile in this figure panel does not seem to capture the general localization of PHF2 seen in the entire image, which might follow that of H3K4me3, consistent with PHF2 binding to TSS, instead of pericentromeric boundaries, which would be rings around chromocenters. The plots in Fig. 4A do not show a striking increase in the first 500kb window that would be much larger than the variations (increase/decrease) in the following 500kb windows. Taking together the data in this paper and in Pappa et al 2019 suggests to me that PHF2 binds mostly to TSS, and only weakly to pericentromeres or their boundaries. This questions the proposed model of PHF2 upregulating satellite transcripts by limiting heterochromatin spreading. The authors should consider the alternative model that PHF2 regulates the genes whose TSS it binds, and that the observed changes in heterochromatin are an indirect downstream effect. I think this model is also consistent with the data presented in the current manuscript, and additional work is required to distinguish between the two possibilities.

Specific major points:

- The authors mention "comprehensive mass spectroscopy ... assays" in the Abstract but do not show any mass spec data in the paper. Rather, they show the interaction of PHF2 with HP1BP3 and SUV39 by Co-IP (Fig. 1B). The authors should show and discuss the mass spec data if they advertise them in this way, and there should be an accession number in the data availability section so that they can be accessed by the reader. Which interaction partners did the mass spec experiment yield, only HP1BP3 and SUV39H1? This is very unlikely, as PHF2 also binds to TSS, where it will colocalize with the transcription machinery. To understand the function of PHF2, it would be particularly helpful to present quantitative mass spec data, in order to estimate if the major PHF2 pool binds to the transcription machinery or to heterochromatin components.

- The analysis in Fig. 1C-E tests the localization of PHF2 at repeat elements but not at pericentromeric boundaries. I do not understand the rationale, as the authors claim later that PHF2 binds these boundaries. The authors should test unique sequences at these boundaries, not only Col2a1 as negative control. Including active TSS in the analysis as a reference would also help, to put this localization at repeats in the context of what was described in Pappa et al, 2019. The open questions for me are: Is the enrichment at repeats stronger/weaker than at TSS? Is there indeed an enrichment at boundaries, and how strong is it compared to the enrichment at satellites and at TSS?

- I do not see the drastic increase in the volume of chromocenters (more than 10x in NIH3T3 according to the boxplot) in the images of Fig. 2C. At first sight, the chromocenters look quite similar to me, just the DAPI "background" might slightly change. The drastic difference in the quantification might come from a problem with the segmentation threshold. If the authors want to

seriously make this claim, I would strongly recommend to perform DNA-FISH against major satellites, which gives higher contrast than DAPI that stains all the chromatin, and to perform the volume analysis on these FISH images. To analyze their DAPI images, the authors could use an intensity histogram for the nucleus of interest to define an appropriate segmentation threshold. If the segmentation threshold is kept constant (which is what I understand was done according to the Methods section), and if the background varies between conditions, for example because the cell density varies, the segmentation result might not be very reliable. Also, I am quite puzzled that the authors obtain different foci numbers and volumes for DAPI and H3K9me3 in wildtype cells. Wouldn't all chromocenters be visible in both channels (in the wildtype condition), and shouldn't the foci number and volume therefore be very similar?

- I find the link to heterochromatin spreading very vague. From the theoretical side, I agree that increased spreading could titrate out heterochromatin proteins, because there are additional binding sites next to the pericentromeres that represent competing binding sites. This would, however, also be true if additional binding sites are formed elsewhere in the genome, for example at TSS where H3K9me3 goes up when PHF2 is removed. These regions could also sequester heterochromatin proteins. I find it therefore not straightforward to invoke spreading as the most likely mechanism at play. In addition, it would be helpful if the authors could support their idea that heterochromatin components are limiting in mammalian cells, so that ectopic H3K9me3 would indeed lead to the activation of normally silenced genes. I am not aware that this has been shown in mammals. From the experimental side, I find the variations shown in Fig. 4A/D quite subtle. Comparing the area under the curve in the first 500kb window to that in the following windows, it looks like PHF2 might be slightly redistributed, but there is no signature that is entirely clear to me. To know the noise level of these curves, it would be important to have error bars on the curves, which come from biological replicates. This would help to distinguish signal from noise. In Fig. 4C, it would be helpful to have a systematic analysis to ask the question if the PHF2 signal is enriched at the chromocenter boundary. Using line profiles, the result will vary depending on which chromocenter and which direction is chosen. The analysis could be done using average radial profiles originating from the centers of all chromocenters, which can easily be done in ImageJ/Fiji (for example with this plugin <https://imagej.net/ij/plugins/radial-profile.html>). Alternatively, PHF2 "foci" and chromocenters could be segmented across the entire nucleus, and their distance distribution could be analysed. I would also strongly suggest to conduct a H3K4me3/PHF2 co-staining and test for co-localization. What we see in these images might be PHF2 sitting at active TSS, not at pericentromeric boundaries.

- I find the analysis of the PHF2 mutants and the effect on satellite transcripts (Fig. 6A/B) very interesting. I am wondering about the localization of the mutant proteins and the H3K9me2/3 distribution in the respective cells. Are there differences in PHF2, H3K9me2/3 at/around chromocenters? Are there differences in the global levels of H3K9me2/3? It would be interesting to have images with higher magnification of the PHF2/DAPI channels (higher than in Fig. 6D) to address these questions, along with a staining of H3K9me2/3. This could give insight into the question if PHF2 regulates transcription in cis or not, and if this occurs via modulation of H3K9 methylation in cis.

- I am not entirely convinced that the model in Fig. 6E describes the findings in this paper very well. PHF2 and SUV39 were shown to interact in Fig. 1, but are separated from each other in the cartoon. There is no data on SUV39 in the paper besides the Co-IP, so that it is largely unclear how SUV39 localizes in the presence/absence of PHF2. There is in my opinion also no strong indication in the paper that SUV39 indeed spreads H3K9me3 out to the next gene as shown in the cartoon. All this could be tested and would probably require inhibition/depletion of SUV39, analysis of the distribution of PHF2/SUV39/H3K9me2/3 at the region between the pericentromere and the next gene (IGV snapshots of the respective regions), at genes ordered according to their distance from the pericentromere, etc. However, in the current state, where such analyses have not been done, this model seems very speculative, and I am not convinced that it can be favored over other models, where depletion of PHF2, which according to the microscopy data in Fig. 4 does not really localize at chromocenters, indirectly induces changes in satellite transcript levels.

Minor points:

- In Fig. 3B, it might be helpful to plot a larger window around the PHF2 peak so that it is clear if there is an enrichment of H3K9me3 around PHF2 (in the form of a colocalizing H3K9me3 "peak", even if broad) or if PHF2 peaks just localize to broad H3K9me3 domains, which is somewhat different

- I do not think that the discussion about LLPS is particularly helpful. The result that the charged domain in the IDR of PHF2 does not have an effect on satellite transcripts does in my opinion not yield much information about the question if PHF2 undergoes LLPS or not at different places in the cell. I would personally limit this part to prevent confusion.

ANSWER TO THE EDITOR AND THE REVIEWERS

EMBOR-2023-58575V1 "PHF2-mediated H3K9me balance orchestrates heterochromatin stability and progenitor proliferation in early neurogenesis"

We sincerely appreciate both the editor's and the reviewers' constructive comments. We have made an effort to provide an answer to all the suggestions. In particular, to improve the quality of the data (more replicates, more controls, validation qPCRs, new immunostaining analyzed by super-resolution microscopy, etc.), to further reinforce the data of PHF2 location at the pericentromeric boundaries (new ChIP assays, immunoassays and new analysis), and to reinterpret our results modulating the conclusions.

We include below a detailed, "point-by-point" response to them. Reviewers' suggestions are printed in Times bold, and our replies are printed in Times. As a guide, however, we outline below the most important changes in the figures of the revised manuscript.

4 new figures (3 supplementary and 1 main) have been added.

MAIN FIGURES:

Figure 1: Panel C contain data from two new PHF2 ChIP-seq replicates. Panel E has been modified, including a TSS as a positive control.

Figure 2: Images in panels C and D have been modified including images with higher resolution and magnification are now shown.

Figure 3: Panel A has been improved showing 5Kb around PHF2 peak. New panels (E and F) have been added, confirming the decrease of H3K9me3 and HP1 α at satellite repeats by ChIP-qPCR. New panel G depicts the decrease of H3K9me3 levels in shPHF2 by *in vivo* imaging.

Figure 4: Panels A and B correspond to panels D and E from former figure 4. Panels C and D and E are new. They display H3K9me3 and HP1 α ChIP-qPCR results around the pericentromeric boundaries and other genomics locations.

Figure 5: Panels A and B correspond to panels A and B from former Figure 4 but they contain data from two new PHF2 ChIP-seq replicates. Panel D depicts PHF2 location by super-resolution microscopy. The rest is new data relative to the transcriptional activity around the pericentromeric repeats.

Figure 6: It corresponds to former Figure 5.

Figure 7: It is the former Figure 6 in which panel B is new (depicts results from rescue experiments in NSCs) and panel E has been eliminated.

SUPPLEMENTARY FIGURES:

Figure EV1: It contains data from the former Figure S1 (A), but data relative to NIH3T3 cells (panels D and E) has been added. Moreover, panel B contains data from two new PHF2 ChIP-seq duplicates. New panel C depicts captures showing the PHF2 binding to some pericentromeric repeats.

Figure EV2: It displays data from the former Figure S4

Figure EV3: It is a new figure showing some genomic features of the pericentromeric boundaries.

Figure EV4: It corresponds to former Figure S6, but panel A from former Figure 7 has been added. Panel C is totally new, showing H3K9me3 foci volume with higher resolution than in the previous version. Images from super-resolution microscopy have been added.

Appendix Figure S1: It corresponds to former Figure S1 in which Panel B has been moved from Fig. 1B. Panels D and E show the quality of the ChIP-seq assays.

Appendix Figure S2: It corresponds to former Figure S2. Panel B has been added showing the levels of PHF2 in shPHF2 in NIH3T3 cells and NSCs.

Appendix Figure S3: It contains data from the former Figure S3, but captures showing the levels of H3K9me3 have been added (B) and ChIP-qPCR to validate the ChIP-seq has been included (C).

Appendix Figure S4: It contains data previously presented in Fig. 4C. New panel A contains data from PHF2 ChIP-qPCR showing the PHF2 binding to boundaries and promoters.

Appendix Figure S5: It displays data from the former Figure S5, but captures displaying the ATAC signals at some regions are included (panel D).

REVIEWER 1

We thank reviewer 1 for their constructive and positive comments. We acknowledge the significance of the raised issues and, in response to their recommendations, we have performed additional experiments to address them and to improve the quality of our data. Specifically, we have made an effort to strengthen the ChIP-seq data, validate our main findings across an alternative cell line, and execute rescues of satellite expression in Neural Stem Cells (NSCs).

1. However, I have concerns about the robustness of the data analyses underlying the conclusions. ChIP-seq repeat-analyses are much challenging and error-prone, among others due to (1) differences in signal-to-noise between individual ChIP-seq experiments;.... The first could be addressed by performing sufficient replicates (including ChIP-qPCR).

Response:

We thank referee 1 for raising this issue. We agree with the referee 1 about the importance of adding more replicates. In response to the referee's recommendation and with the intent of enhancing the robustness of our dataset, we performed new PHF2 ChIP-seq experiments per duplicated, obtaining similar results to those presented in the first version of our manuscript. The results are now displayed in the Fig. 1C, Fig. EV1B,C, and Fig.5A-C. Moreover, the quality of the experiments has been assessed, see response 3.

2. Potential differences in abundance of the ChIPped protein. The first could be addressed by performing sufficient replicates (including ChIP-qPCR), the latter using spike-in ChIP-seq.

Response:

We agree with referee 1 about the difficulty to compare ChIP-seq signal. Although we conducted these experiments simultaneously, they have been normalized to the inputs, and the same number of reads were sequenced. Spiking is particularly advised when starting from cells with differing chromatin content, which is not applicable in our scenario and the use of spikes is currently quite controversial. To prove that epigenetic instability was associated with a H3K9me3 decrease at heterochromatin regions containing major satellites, we utilized modular fluorescence complementation sensors, A technique that does not require cell fixation or the use of antibodies. These sensors allowed us to detect H3K9me3 signals specifically at the MajSat genomic site in live cells, as illustrated in Figure 3G (Lungu et al., 2017). Live-cell imaging of both shCT and PHF2-depleted cells revealed a decrease in H3K9me3 levels at major satellite regions compared to the remaining cellular levels (Figure 3G).

Additionally, to reinforce our data on H3K9me3, we have validated the ChIP-seq presented by qPCR (Fig. 3E and Appendix Fig. S3C) and we have performed new ChIP-qPCR for HP1a. The results have shown that just as H3K9me3 decreases in satellites (Fig. 3E), so does HP1a (Fig. 3F). The opposite occurs in the boundary regions where both H3K9me3 and HP1a increase (Fig. 4C and D).

We have analyzed the H3K9me3 levels of centromeres in mitotic chromosomes separated by FACS through immunostaining as we proposed in the revision plan, obtaining similar results; however, we have not included these data, because as referee 3 suggested, mitotic chromosomes might not reflect the status of interphase chromatin.

3. It would be helpful if additional data is provided showing the quality and success of the individual ChIP-seq experiments, in particular by showing genome browser views showing the quality of the data, and relevant non-binned changes at the loci under investigation.

Response:

In accordance with the recommendations of Referee 1, we have assessed the quality of the ChIP-seq experiments. To enhance the robustness of our study, we have incorporated novel quality controls:

- 1.- qPCR analyses to validate the H3K9me3 and ChIP-seq in control and PHF2 KD cells (Appendix Fig S3C)
- 2.- we have provided a table containing the read count illustrating ChIP-seq mapping statistics for all ChIP-seq data presented in the manuscript. (Appendix Table S2).
- 3.- we conducted a correlation analysis (Pearson correlation and PCA) to demonstrate that the replicates from various samples are appropriately clustered (Appendix Fig S1D). As can be observed, the correlation between the replicates shown in the previous version is slightly lower than that observed with the two conducted during the manuscript review. Probably because the four duplicates were not performed in parallel and the batch of the antibody used is different. However, the results reproduce and reinforce the initial conclusions (Fig. 1C, Fig. EV1B, Fig. 5A-C).
- 4.- In response to the reviewer's specific recommendation, we have also generated IGV genome browser screenshots illustrating the continuous quantification of our ChIP-seq samples within specific regions of interest (Figure EV1C, Appendix Fig S1E, S3B, S5D).

4. The above are even more important as the ChIP-qPCR presented by the authors show extremely low recovery percentages (below 0.1% for positive loci, Fig 1e/ S1b). This is important as the other main read-out of the authors represent IF. And while the authors generally provide sufficient statistics for their IF experiments, also IF is prone to high variability (e.g. variation between replicates and cell cultures; and variation within slides).

Response:

We acknowledge the referee's observation that ChIP-qPCR values are low, although they are both reproducible and statistically significant. Comparable values have been reported (Pappa et al., 2019; Bricambert et al., 2018; Kim et al. 2019) for other regions, such as promoters. It is important to note that the percentage of enrichment relative to the input is heavily influenced by the proximity of the chromatin to the targeted chipped factor (transcription factor, histone, co-factor) and the antibody's affinity. At the time of the revision, we obtained a new batch of PHF2 antibody with which we repeated the ChIP-seq experiments. Upon observing better enrichments, we repeated the ChIP-qPCR experiments, which yielded higher % input for both promoters and satellites. The new data are in Fig. 1E, Appendix Fig. S4A and Appendix Fig. S1C (bottom panels).

Finally, it is noteworthy that in the immunofluorescence experiments, only those conducted in parallel, with images obtained using the same settings, have been compared. These experiments include multiple replicates (yielding consistent results in all cases), and the quantification involves more than 100 cells in every instance. The conclusions drawn are always based on results obtained from multiple techniques, never relying on a single method alone.

5. Also, the authors should validate their main findings in another cell line (potentially NIH3T3 which is already used in few experiments).

Response:

In response to the reviewer's recommendation, we replicated the main analysis of our study utilizing two ChIP-seq replicates of PHF2 in the NIH3T3 cell line, where we had previously demonstrated the activation of repeats upon PHF2 depletion. The outcomes were generally consistent with those previously observed in NSCs, reinforcing our conclusions and validating the principal findings of our investigation (Fig. EV1D,E).

Additionally, we have verified repeat activation in a human cell line (HEK293T), distinct from the previous murine models. These findings are incorporated into Referee's Figure 1.

6. The authors should present the complete experiment with all PHF2 interactors, and interpret these data in the context of their paper.

Response:

At the referee's suggestion, we have deposited the data and presented the entire interactome in the Appendix Table S1. The data do not suggest interaction with any complete pathways but do reveal specific targets related to RNA processing (RL14, RS8), transcription (SSRP1 (FACT complex), BTF3), three-dimensional chromatin structure (Lamin B and SMC2), DNA repair (YBX1), and heterochromatin (HP1BP3). We specifically discuss and focus on HP1BP3 because clear clusters suggesting specific involvement in a process are not observed. Identification of single interacting partner for PHF2 has been described by other (Bricambert et al., 2018; Kim et al. 2019). These results are discussed in the new version of the manuscript.

7. The authors mainly rely on NSCs (sometimes in combination with NIH3T3). But for the rescue experiments in Fig 6b, the authors only use NIH3T3 for unclear reasons.

Response:

Referee 1 is correct; The rescue experiments involving the transcription of satellite repeats were conducted in NIH3T3. However, rescues addressing damage and genomic instability, indicative of increased satellite activity, were performed in NSCs. The technical limitation stems from the fact that NSCs cannot be transfected directly; instead, they are manipulated through lentiviral infection or nucleofection, the latter yielding a low but sufficient percentage for single-cell analysis via immunostaining. Challenges have arisen with the expression of one mutant in lentiviral vectors. In response to Referee 1's suggestion, we have now cloned the mutants into a different lentiviral vector pLIP and performed the rescue of satellite transcription in both NSCs and NIH3T3. The results validate the data presented for DNA damage, wherein both PHF2 WT and the charged mutant rescued satellite expression, while the PHD and catalytic mutant did not. These new results are now depicted in Fig. 7B.

8. A better characterization of the PHF2-associated boundaries would be helpful. I feel it is important to show what happens to gene expression at these boundaries, data which the authors refer to but excluded ("data not shown").

Response:

We thank referee 1 for raising this intriguing question. Boundary regions are highly interesting areas, sparsely populated with genes, characterized by small tracks of repetitive DNA, and relatively understudied. In an effort to further characterize these zones and determine their potential relationship with the presence of PHF2, we have conducted the following analyses:

(i) We examined changes in expression within the first Mb from pericentromeric repeats following PHF2 depletion. Unlike the previous analysis, in which we analyzed only mRNA, this time we have analyzed total changes in RNA detected in RNA.seq. Simultaneously, we compared their expression with levels of RNA Pol II. Our analysis revealed a decrease in transcription levels in the pericentromeric regions, particularly between 3-3,5 Mb from the satellite (Fig. 5E). Interestingly, these regions, exhibit a high level of PolII binding (Fig. 5F), particularly evident in certain chromosomes such as Chr 19 and 11 and low levels of histone H1 (Fig. EV3A). This finding suggests that, it is likely that disruption in methylation levels in the boundary region leads to transcriptional decrease that could potentially contribute to the instability of pericentromeric repeats. An alternative explanation to the transcriptional changes observed at the boundaries could be that they are not the cause but rather the consequence. Therefore, the presence of a demethylase that limits the effective local levels of H3K9me3 could potentially restrict its spreading, as previously suggested (Aygün *et al.*, 2013; Ragunathan *et al.*, 2015). This demethylation allows for the maintenance of high levels of H3K9me3 within the repeats.

Consequently, correct levels of H3K9me3 help to balance the components necessary for the write/read system, which are required for the formation of the H3K9me3 hub responsible for heterochromatin stability.

(ii) We examined other genomic features of these regions, such as G/C content. However, we did not find any significant bias in these regions.

(iii) Moreover, it is well established that genes for noncoding transfer RNAs (tRNAs) function as boundary elements (Ebersole et al., 2011; Raab et al., 2012), thus, we investigated the presence of tRNAs and ncRNAs and observed an enrichment of these genomic elements across the boundary regions in certain chromosomes (Fig. EV3B,C). However, we have been unable to assess the potential role of PHF2 in their expression due to our inability to detect them in the RNA-seq data.

Collectively, the results suggest that PHF2 plays a role in balancing H3K9me3 levels. This is essential for stabilizing PcH either by probably regulating transcription at the boundaries, and/or disrupting the distribution of heterochromatin components.

9. There is no spread of samples on the PC1 in Fig S4b; As such, this figure as it is now does not make sense.

Response:

The referee is totally right. The figure has been eliminated.

10. For the ChIP-seq repeat analysis Figs 1d, S1c, 4b and 4e, it is unclear which exact values the y-axis represent.

Response:

The mean ChIP-seq signal of each of the samples within the repetitive sequences included in the groups of repeats of interest was initially computed, and the boxplots display the distribution of these values. The term "ChIP-seq signal" refers to the samples' CPM-normalized and input-subtracted read count. We acknowledge the reviewer's observation regarding the ambiguity of this information in the figures mentioned. Consequently, we have incorporated this clarification into the respective figure legends.

11. The proteome data (Fig 2b) does not seem to be submitted to a public database.

Response:

It has been deposited.

REFEREE 2

We express our gratitude to Reviewer 2 for providing constructive and positive feedback. Recognizing the importance of the raised concerns, we have undertaken additional experiments and include more explanations in response to their suggestions.

1. I do not understand the point of talking about genome stability before talking about repeated sequences that can recombined (Line 5 of the introduction).

Response:

We apologize for the lack of clarity in describing genome stability. Following their suggestion, we have changed the position of this sentence in the introduction. Indeed, a significant portion of the introduction has been rewritten to provide a clearer understanding of the rationale behind our utilization of NSCs, PHF2, and its potential association with heterochromatin. For detailed insights, kindly refer to response 2.

2. The neural stem cell (NSC) model and the focus on PHF2 protein are introduced a little too abruptly for me... Why NSC and not ESC? Why PHF2 and not a histone methyltransferase like G9a or Setdb1 (which are not mentioned as actors of the methylation of H3K9) or another histone demethylase? Please explain a little more or differently.

I need more information about the relevance of the cellular model and the focus on this enzyme. Also, I am not in favor of the use of NIH3T3 cells line (a cell line with an abnormal karyotype and genome instability) to study heterochromatin stability...

Response:

We fully agree with the referee that the introduction of PHF2 and NSCs was abrupt in the previous version of our manuscript. To better address these considerations and to clearly introduce why we use NSCs and PHF2 to study heterochromatin, a significant part of the introduction has been rewritten. We hope that in the new version, these aspects have been clarified.

As the referee indicates in their subsequent comment, this article is a follow-up to our previous study on PHF2 in the laboratory (Pappa et al., 2019). In this work (Pappa et al, 2019), we focused on PHF2 in the context of neurodevelopment because PHF2 was initially identified as a candidate gene for hereditary sensory neuropathy type I (Nicholson et al., 1996), it is expressed at high levels in the neural tube and dorsal root ganglia (Hasenpusch et al., 1999), and has been frequently associated with autism and other neurodevelopmental disorders. In Pappa et al, we found that PHF2 is essential for NSCs proliferation both in vitro and in vivo. Intriguingly, PHF2 depletion induces R-loop accumulation, leading to extensive DNA damage in NSCs. At that time, we did not identify the molecular mechanisms involved in this process. The formation of R-loops and

DNA damage often occurs when transcription and replication machineries collide, suggesting that heterochromatin might be destabilized, transcribing out of sync and generating these collisions between molecular machineries. This was the starting point for the use of NSCs and PHF2.

Finally, we utilized well-characterized E12.5 mouse NSCs derived from the cortex of E12.5 mouse embryos. These cells are maintained for several passages in culture and retain their stemness capacity. They proliferate and can generate wide range of differentiated neural cell types (Currle et al., 2007; Pollard et al., 2006; Theus et al., 2012). All these considerations have been included in the new version of the manuscript in the Introduction and Results sections (pages 3-5).

Regarding the use of NIH3T3 cells, we wholeheartedly agree with the referee. It is not the ideal model for studying heterochromatin destabilization or DNA damage because they already possess these characteristics in control conditions (although it increases after PHF2 depletion). We used these cells occasionally to confirm that the role of PHF2 could be general and common to different cell models, although the major conclusions were drawn using NSCs

3. I guess this study is a follow-up of the previous study of the authors (Pappa et al 2019) but readers still need more explanation to understand the purpose of this study and why it is relevant to study H3K9me2/me3 and PcH transcription/organization in the context of a histone H3K4 demethylase.

Response:

Thank you for this comment that allows us to clarify the main objective of our study, which may not have been entirely clear in the introduction of our manuscript. PHF2 is a demethylase for H3K9me1/2, as demonstrated in our previous article, maintains low levels of H3K9me2/3 at some promoters. We now shift our focus to heterochromatin, as the repression of heterochromatin is mediated by specific H3K9me2/3. This histone mark is deposited by suppressor of variegation 3-9 homolog 1/2 (SUV39H1/H2) and recognized by HP1 (Grewal, 2023; Martens et al., 2005), facilitating their silencing through a read-write mechanism mediated by the chromodomain of HP1. The density of H3K9me3 is crucial to support the read-write mechanism for heterochromatin spreading. Therefore, the presence of a demethylase might limit the effective local levels of H3K9me3, potentially restricting its spreading, as previously suggested (Aygün et al., 2013; Ragnathan et al., 2015).

Based on this premise, we decided to investigate the role of the histone demethylase activity of PHF2 in heterochromatin. All these considerations are now included in the new introduction of the revised version (pages 3-4).

4. The authors should present more openly the data of ChiP analysis for H3K9me2 it's confusing in the text because they only talk clearly about H3K9me3 and H3K4me3?

Response:

We appreciate the referee's question, as it allows us to clarify why we opted for H3K9me3 and H3K4me3 instead of H3K9me2, which is the target of PHF2. To address this point, we would like to make three considerations:

1. The primary reason for monitoring H3K9me3 levels is that it serves as a marker for constitutive heterochromatin, that is the target of this study.
2. On the other hand, although we observed an increase in H3K9me2 levels in NSCs after depleting PHF2, this increase is more pronounced for H3K9me3 (Pappa et al., 2019). This may be because SUV39H utilizes H3K9me2 as a substrate, resulting in a noticeable rise in H3K9me3 levels.
- 3.- H3K4me3 was chosen as it has been reported that H3K9me3-marked heterochromatin domains are delimited by the presence of H3K4me3. So, upon observing changes in H3K9me3 levels and extension, we wondered if it affected the extension of H3K4me3 domains. Therefore, we included the ChIP-seq analysis of H3K4me3
- 4.- Finally, we used already published H3K9me2 ChIP-seq to compare H3K9me2 with the presence of PHF2. We observed that both signals are mutually exclusive (Figure 1D).

6. By the way, were the H3K9me3 and H3K4me3 data already published? If yes it should be explicitly written.

Response:

We appreciate the referee's inquiry, as it provides an opportunity to clarify that the ChIP-seq experiments were conducted in-house in NSCs and are deposited in the GEO database under the accession GSE242383 (access token yxwnumeixlubvgz). The obtained results were also cross-referenced with publicly available ChIP-seq data in NSCs, yielding consistent findings. The motivation behind conducting H3K9me3 and H3K4me3 ChIP-seq experiments was to investigate the impact of PHF2 depletion on these histone marks. Consequently, these experiments were carried out in both control and PHF2-depleted NSCs.

This clarification is now provided on page 7.

7. Page 8 last sentence: A reference is needed about the definition of chromocenters in mouse

Response:

Based on the referee's suggestion, we have incorporated the definition of chromocenter as "prominent DNA-dense nuclear foci that contain clustered pericentromeric satellite DNA repeats from multiple chromosomes in interphase mouse cells (Pardue&Gall, 1970; Jones 1970)" see in page 6.

8. In Fig S6B It would be more understandable to add representative images of the labeling (H3K9me3) of chromocenters like in Fig2.

Response:

The images of H3K9me3 have been included in the new version of Fig. EV4C, as referee suggested.

9. More explanation would be needed on the various degrees of alteration in the chromocenters foci.

Response: We would like to apologize for the sentence “mutants generate various degrees of alteration in the chromocenters foci” which was added in error in the previous version of the manuscript. To clarify the role of the mutants not only in inducing repeat transcription (Fig. 7B and Fig. EV4A) but also in the volume of the foci, we performed rescue experiments by expressing the different mutants in PHF2 depleted cells. According to the rescue results in satellite expression and DNA damage (Fig. 7B,C, Fig. EV4A,D) the catalytic and PHD mutants did not rescue the volume of the foci, but PHF2 WT and the charged mutant did. These new results are shown in Fig. EV4C of the new version of the manuscript, and the mistake has been corrected.

REFEREE 3

I would like to express my gratitude to referee 3 for their constructive comments, which have prompted us to reflect on certain aspects of our work, especially the interpretation of the results. I am confident that these comments have contributed to the overall improvement of our work. Following their guidance, we have made an effort to analyze in detail the regions of the boundaries and the potential role of PHF2 in this area that could directly or indirectly affect satellite transcription. We have conducted genomic analysis (G/C content, enrichment of tRNAs and ncRNAs), as well as examined changes in the transcription after PHF2 depletion, which could impact heterochromatin stability. Additionally, we have performed qPCRs to confirm PHF2 localization and used super-resolution microscopy to improve the resolution and quality of some images. Based on the results obtained and with the guidance provided by the referee, we have modified the interpretation of our results, especially concerning heterochromatin spreading.

Lastly, we have added numerous controls to strengthen the robustness of our data

1. The authors should show and discuss the mass spec data if they advertise them in this way, and there should be an accession number in the data availability section so that they can be accessed by the reader.

Response:

We value the referee's suggestion. As described in response 6 from referee 1, the data has not been deposited because the manuscript's objective was not to showcase the PHF2 proteome but to demonstrate PHF2's interaction with a specific heterochromatin protein. The suggestion that PHF2 might interact with HP1BP3, confirmed by CoIP, emerged from a single replica of mass spectrometry analysis. We confirm HP1BP3, and this result prompted further exploration in heterochromatin. Our intention is solely to illustrate the interaction between these two factors, not to conduct a comprehensive study determining the entire PHF2 proteome. After consulting with the editor, we have deposited the data and presented the entire interactome in Appendix Table 1. The data do not suggest interaction with any complete pathways but do reveal specific targets related to RNA processing (RL14, RL29, RS8), transcription (SSRP1 (FACT complex), BTF3), three-dimensional chromatin structure (Lamin B and SMC2), DNA repair (YBX1), and heterochromatin (HP1BP3). The accession number is now included in the revised version of the manuscript in Data Availability.

2. The authors should test unique sequences at these boundaries, not only Col2a1 as negative control. Including active TSS in the analysis as a reference would also help, to put this localization at repeats in the context of what was described in Pappa et al, 2019. The open questions for me are: Is the enrichment at repeats stronger/weaker than at TSS? Is there indeed an enrichment at boundaries, and how strong is it compared to the enrichment at satellites and at TSS?

Response:

We appreciate the referee for this suggestion, which allows us to compare the binding of PHF2. Following their recommendations, new qPCR experiments have been conducted using previously identified TSS that are direct targets of PHF2, as well as boundary regions of pericentromeric heterochromatin located between 3-4 MB from the centromeric end of the chromosome, along with satellites.

The results reveal that PHF2's affinity for pericentromeres is lower than that for TSS and boundaries. The latter are of the same order of magnitude and, depending on the analyzed promoter, lower than the TSS of p21 and similar or higher than Brca2, well-described targets of PHF2. These findings suggest that although PHF2 binds significantly to pericentromeres, they are not the sites of highest affinity in the genome as expected; instead, these are the TSS and boundaries. These findings are depicted in Figure 1E and Appendix Figure S4A.

3. I do not see the drastic increase in the volume of chromocenters (more than 10x in NIH3T3 according to the boxplot) in the images of Fig. 2C....

To analyze their DAPI images, the authors could use an intensity histogram for the nucleus of interest to define an appropriate segmentation threshold. If the segmentation threshold is kept constant (which is what I understand was done according to the Methods section), and if the background varies between conditions, for example because the cell density varies, the segmentation result might not be very reliable. Also, I am quite puzzled that the authors obtain different foci numbers and volumes for DAPI and H3K9me3 in wildtype cells. Wouldn't all chromocenters be visible in both channels (in the wildtype condition), and shouldn't the foci number and volume therefore be very similar?

Response:

We agree with the referee's assessment. The DAPI images were not of sufficient quality for precise quantifications. Consequently, we have repeated the experiments, obtaining improved images with higher magnification and resolution, and in some cases, we have used super-resolution microscopy. Upon obtaining these images, we have followed the suggestions provided by referee 3. The updated quantifications are presented in Figure 2C,D and Fig. EV4C, in alignment with the rest of the findings, demonstrate a more moderate increase in the volume of foci compared to what was previously shown

4. I find the link to heterochromatin spreading very vague. From the theoretical side, I agree that increased spreading could titrate out heterochromatin proteins, because there are additional binding sites next to the pericentromeres that represent competing binding sites. This would, however, also be true if additional binding sites are formed elsewhere in the genome, for example at TSS where H3K9me3 goes up when PHF2 is removed. These regions could also sequester heterochromatin proteins. I find it therefore not straightforward to invoke spreading as the most likely mechanism at play. In addition, it would be helpful if the authors could support their idea that heterochromatin components are limiting in

mammalian cells, so that ectopic H3K9me3 would indeed lead to the activation of normally silenced genes. I am not aware that this has been shown in mammals.

Response:

Following referee's suggestion, we have explored the possibility of other mechanisms, such as transcriptional changes at boundaries or increased H3K9me3 in other locations different than boundaries (as suggested by the referee) could contribute directly or indirectly to heterochromatin transcription.

We have performed the following analysis:

1.- We have conducted experiments to elucidate the contribution of increase of H3K9me3 in other genomic regions (different than boundaries) to the dilution of the heterochromatin components. This involved identifying upregulated regions in PHF2 knockdown (KD) that exhibit H3K9me3 under control conditions. Next, we have investigated the binding of heterochromatin components, such as HP1a, at these regions through ChIP-qPCR assays. The results revealed a decrease in its recruitment to analyzed heterochromatic loci upon PHF2 depletion, while an increase was observed over the boundary regions (Fig. 4D,E). This observation suggests a potential redistribution of HP1a, and likely other heterochromatin components, upon PHF2 depletion.

2.- In addition to the dilution of heterochromatin components due to heterochromatinization of regions other than boundaries, we have explored the possibility that transcriptional changes which have been described as regulators of heterochromatin boundaries (Lunyak *et al*, 2007; Raab *et al*, 2012; Scott *et al*, 2006) could affect heterochromatin stability. To this end, we have thoroughly analyzed boundaries in control and PHF2 KD for changes in transcription. Unlike the previous analysis, in which we analyzed only mRNA, this time we have analyzed total changes in RNA detected in RNA-seq. Simultaneously, we compared their expression with levels of RNA Pol II ChIP-seq. Our analysis revealed a decrease in transcription levels in the pericentromeric regions, particularly between 3-4 Mb from the satellite (Fig. 5E). This finding suggests that, it is likely that disruption in methylation levels in the boundary region leads to transcriptional decrease that could potentially contribute to the instability of pericentromeric repeats.

3.- Finally, in order to understand the contribution of boundaries regions to the described PHF2 effects on Pc heterochromatin we have thoroughly analyzed boundaries regions for the presence of genomic motifs: enrichment of tRNA genes, high G/C content, histone H1 content, etc. We have observed an enrichment of ncRNA and tRNA across the boundary regions in certain chromosomes (Fig. EV3B,C). However, we have been unable to assess the potential role of PHF2 in their expression due to our inability to detect them in the RNA-seq data.

Collectively, we have explored the possibility of other mechanisms, such as transcriptional changes or increased methylation in other locations (as pointed by the referee), and the results suggest that indeed alternative pathways to the one proposed in our initial version, involving heterochromatin extension, are possible to maintain heterochromatic stability. These new interpretations have been incorporated and discussed in the

revised manuscript. It is important to note that the interpretation has been modified but remains compatible with the presented data and the overall conclusion that PHF2 is crucial for maintaining silenced heterochromatin and genetic stability, requiring its catalytic domains and PHD.

5.- From the experimental side, I find the variations shown in Fig. 4A/D quite subtle. Comparing the area under the curve in the first 500kb window to that in the following windows, it looks like PHF2 might be slightly redistributed, but there is no signature that is entirely clear to me. To know the noise level of these curves, it would be important to have error bars on the curves, which come from biological replicates. This would help to distinguish signal from noise

Response:

To address this question, we computed the standard error obtained from the two PHF2 biological replicates in each window. Intending to represent the standard error as cleanly as possible, we added two additional line plots (smoothed) illustrating the trend of this measure. These correspond to the lighter lines surrounding the main one. This approach demonstrates that the enrichment is not attributable to noise, as the standard error is relatively low and remains nearly constant throughout the entire length of the pericentromeric regions across all chromosomes.

6.- In Fig. 4C, it would be helpful to have a systematic analysis to ask the question if the PHF2 signal is enriched at the chromocenter boundary. Using line profiles, the result will vary depending on which chromocenter and which direction is chosen. The analysis could be done using average radial profiles originating from the centers of all chromocenters, which can easily be done in ImageJ/Fiji (for example with this plugin <https://imagej.net/ij/plugins/radial-profile.html>). Alternatively, PHF2 "foci" and chromocenters could be segmented across the entire nucleus, and their distance distribution could be analysed.

Response:

To enhance our analysis, we incorporated new images obtained through super-resolution microscopy. For the analysis we have used radial profiles as recommended by referee 2. The updated images and analysis are shown in Fig. 5D

7.- I would also strongly suggest to conduct a H3K4me3/PHF2 co-staining and test for co-localization. What we see in these images might be PHF2 sitting at active TSS, not at pericentromeric boundaries.

Response:

We have previously demonstrated (Papa et al., 2019) the colocalization of PHF2 with H3K4me3 at transcription start sites (TSS) in NSCs through ChIP-seq experiments. However, the profiles obtained around the pericentromeres for PHF2 and H3K4me3 in this region do not coincide. In fact, they are nearly mutually

exclusive, particularly evident within the first 500 Kb next to the centromere (3-3.5 Mb) (see Figure 5A and Figure EV2). Consequently, we believe that the genomic localization observed around the pericentromeres is not due to binding with H3K4me3.

Nevertheless, following the referee's recommendations, we conducted co-immunostaining using GFP fused to PHF2 (since both the PHF2 and H3K4me antibodies were generated in rabbit). As expected from the genomic data (Pappa et al., 2019), extensive colocalization between GFP-PHF2 and H3K4me3 is detected. However, there are unique binding sites for GFP-PHF2, many of which are located around the pericentromeres. Nevertheless, we believe that this might not be the optimal experiment to observe localization, given the overexpression of GFP-PHF2 and the lack of resolution (see Referee's Fig. 2).

8. I find the analysis of the PHF2 mutants and the effect on satellite transcripts (Fig. 6A/B) very interesting. I am wondering about the localization of the mutant proteins and the H3K9me2/3 distribution in the respective cells. Are there differences in PHF2, H3K9me2/3 at/around chromocenters? Are there differences in the global levels of H3K9me2/3? It would be interesting to have images with higher magnification of the PHF2/DAPI channels (higher than in Fig. 6D) to address these questions, along with a staining of H3K9me2/3

Response:

To clarify the role of the mutants we performed rescue experiments by expressing the different mutants in PHF2 depleted cells. We have repeated the immunostaining assays, now including the H3K9me3 antibody. We made efforts to capture images with higher magnification and resolution, using super-resolution microscopy and analyzing the H3K9me3 foci. According to the rescue results in satellite expression and DNA damage (Fig. 7B,C, Fig. EV4A,D) the catalytic and PHD mutants did not rescue the volume of the foci, but PHF2 WT and the charged mutant did. Moreover, the catalytic and PHD mutants expression led to a global increase in H3K9me3 levels, mirroring the effects observed upon PHF2 depletion (Referee's Fig. 3). These new results are shown in Fig. EV4C of the new version of the manuscript.

9. I am not entirely convinced that the model in Fig. 6E describes the findings in this paper very well. this model seems very speculative, and I am not convinced that it can be favored over other models, where depletion of PHF2, which according to the microscopy data in Fig. 4 does not really localize at chromocenters, indirectly induces changes in satellite transcript levels.

Response:

Referee 3 is absolutely correct; the model lacks accuracy. As the referee suggested alternative interpretations, both direct and indirect, are possible. Consequently, we have chosen to address these possibilities in the discussion section and have opted to remove the model

10. In Fig. 3B, it might be helpful to plot a larger window around the PHF2 peak so that it is clear if there is an enrichment of H3K9me3 around PHF2 (in the form of a colocalizing H3K9me3 "peak", even if broad) or if PHF2 peaks just localize to broad H3K9me3 domains, which is somewhat different

Response:

According to the reviewer's suggestion, we replicated the line plot represented in Fig. 3B (now Fig. 3A) including a larger window around PHF2 peaks by considering adjacent regions of 5 kb. In addition, we opted to employ the computeMatrix and plotProfile functions included in the deepTools suite, as we believe these tools offer enhanced precision for such analyses. The resultant graphs reveal an enrichment of H3K9me3 in the form of a colocalizing broad peak, instead of a broad domain within which the PHF2 peak is situated.

11. I do not think that the discussion about LLPS is particularly helpful. The result that the charged domain in the IDR of PHF2 does not have an effect on satellite transcripts does in my opinion not yield much information about the question if PHF2 undergoes LLPS or not at different places in the cell. I would personally limit this part to prevent confusion.

Response:

Following referee suggestion, the discussion about LLPS has been eliminated

REFERENCES

Aygun O, Mehta S, Grewal SI (2013) HDAC-mediated suppression of histone turnover promotes epigenetic stability of heterochromatin. *Nat Struct Mol Biol* 20: 547-554

Bricambert, M. C. Alves-Guerra, P. Esteves, C. Prip-Buus, J. Bertrand-Michel, H. Guillou, et al. Nat Commun (2018) The histone demethylase Phf2 acts as a molecular checkpoint to prevent NAFLD progression during obesity. Vol. 9 Issue 1 Pages 2092

Currle, D.S., Hu, J.S., Kolski-Andreaco, A. and Monuki, E.S. (2007) Culture of mouse neural stem cell precursors. *J Vis Exp*, 152.

Ebersole T, Kim JH, Samoshkin A, Kouprina N, Pavlicek A, White RJ, Larionov V (2011) tRNA genes protect a reporter gene from epigenetic silencing in mouse cells. *Cell Cycle* 10: 2779-2791

Grewal SIS (2023) The molecular basis of heterochromatin assembly and epigenetic inheritance. *Mol Cell* 83: 1767-1785

Hasenpusch-Theil, K. et al. (1999) PHF2, a novel PHD finger gene located on human chromosome 9q22. *Mamm Genome* 10, 294-8.

Jones KW. (1970) Chromosomal and nuclear location of mouse satellite DNA in individual cells. *Nature* Vol. 225 Issue 5236: 912-5.

- Kim HJ, Hur SW, Park JB, Seo J, Shin JJ, Kim SY, Kim MH, Han DH, Park JW, Park JM *et al* (2019) Histone demethylase PHF2 activates CREB and promotes memory consolidation. *EMBO Rep* 20: e45907
- Lungu C, Pinter S, Broche J, Rathert P, Jeltsch A (2017) Modular fluorescence complementation sensors for live cell detection of epigenetic signals at endogenous genomic sites. *Nat Commun* 8: 649
- Lunyak VV, Prefontaine GG, Nunez E, Cramer T, Ju BG, Ohgi KA, Hutt K, Roy R, Garcia-Diaz A, Zhu X *et al* (2007) Developmentally regulated activation of a SINE B2 repeat as a domain boundary in organogenesis. *Science* 317: 248-251
- Martens JH, O'Sullivan RJ, Braunschweig U, Opravil S, Radolf M, Steinlein P, Jenuwein T (2005) The profile of repeat-associated histone lysine methylation states in the mouse epigenome. *EMBO J* 24: 800-812
- Nicholson, G.A. *et al.* (1996) The gene for hereditary sensory neuropathy type I (HSN-I) maps to chromosome 9q22.1-q22.3. *Nat Genet* 13, 101-4.
- Pappa S, Padilla N, Iacobucci S, Vicioso M, Alvarez de la Campa E, Navarro C, Marcos E, de la Cruz X, Martinez-Balbas MA (2019) PHF2 histone demethylase prevents DNA damage and genome instability by controlling cell cycle progression of neural progenitors. *Proc Natl Acad Sci U S A* 116: 19464-19473
- Pardue ML. and Gall JG. (1970) Chromosomal localization of mouse satellite DNA. *Science* Vol. 168 Issue 3937:1356-8.
- Pollard, S.M., Conti, L., Sun, Y., Goffredo, D. and Smith, A. (2006) Adherent neural stem (NS) cells from fetal and adult forebrain. *Cereb Cortex*, 16 Suppl 1, i112-120.
- Raab JR, Chiu J, Zhu J, Katzman S, Kurukuti S, Wade PA, Haussler D, Kamakaka RT (2012) Human tRNA genes function as chromatin insulators. *EMBO J* 31: 330-350
- Ragunathan K, Jih G, Moazed D (2015) Epigenetic inheritance uncoupled from sequence-specific recruitment. *Science* 348: 1258699
- Ragunathan K (2020) An H3K9 methylation-dependent protein interaction regulates the non-enzymatic functions of a putative histone demethylase. *Elife* 9
- Scott KC, Merrett SL, Willard HF (2006) A heterochromatin barrier partitions the fission yeast centromere into discrete chromatin domains. *Curr Biol* 16: 119-129
- Theus, M.H., Ricard, J. and Liebl, D.J. (2012) Reproducible expansion and characterization of mouse neural stem/progenitor cells in adherent cultures derived from the adult subventricular zone. *Curr Protoc Stem Cell Biol*, Chapter 2, Unit 2D 8

Referee's Figure 1

Figure for referee with unpublished data and its description has been removed upon request by the authors.

Referee's Figure 2

Figure for referee with unpublished data and its description has been removed upon request by the authors.

Referee's Figure 3

Figure for referee with unpublished data and its description has been removed upon request by the authors.

Dear Dr. Martínez-Balbas,

Thank you for the submission of your revised manuscript. We have now received the reports from the referees as well as cross-comments, all pasted below. As you will see, referee 1 still has a few more suggestions that I would like you to address before we can proceed with the official acceptance of your manuscript. Both referees 2 and 3 have commented on referee 1's report, and I suggest that you follow their suggestions for how to address the final concerns. Please let me know if you have any questions, and we can discuss this further.

Please co-submit a point-by-point response with your final submission.

A few editorial requests will also need to be addressed:

- Our routine image analysis shows that the Western blot is re-used in Figure 1A and Appendix Figure S1A. Please explain, and if this is the exact same experiment, please mention this in the Appendix Fig S1A figure legend.
- Please rename the conflict of interest subheading to "Disclosure and Competing Interests Statement"
- Please correct the author names to first name last name in the ms file. It is currently the opposite.
- Please remove the statement DATA NOT SHOWN on p13 and p44 as per journal policy. Please either show the data or re-write.
- In the APPENDIX FILE, each figure legend should follow its figure. Please correct.
- Please note that the box plots need to be defined in terms of minima, maxima, centre, bounds of box and whiskers, and percentile in the legends of figures 1d; 3a, c, g; 4b; 5c, e-f; 6a, e, g; EV 1a-b, e; EV 3a.
- Please note that $n=2$ in figures 3f; 4d-e. In this case no statistics should be calculated. Please remove the error bars and show the single data points of both experiments instead, along with their mean.
- Please note that the measure of center for the error bars needs to be defined in the legends of figure 7b; EV 4a-b.
- Please provide specific URLs for GSE242385, GSE242383, GSE242384, and PXD051336 datasets in the data availability statement.
- Please note that the legends for figures 2c-d is incorrectly labelled as 2c. This needs to be rectified.
- Please note that in figure 2a; there is a mismatch between the annotated p values in the figure legend and the annotated p values in the figure file that should be corrected.

I would like to suggest some minor changes to the abstract that needs to be written in present tense. Please let me know whether you agree with the following:

Heterochromatin stability is crucial for progenitor proliferation during early neurogenesis. It relies on the maintenance of local hubs of H3K9me. However, our understanding of the formation of efficient localized levels of H3K9me remains limited. To address this question, we used neural stem cells to analyze the function of the H3K9me2 demethylase PHF2, which is crucial for progenitor proliferation. Through mass-spectroscopy and genome-wide assays, we show that PHF2 interacts with heterochromatin components and is enriched at pericentromeric heterochromatin (PcH) boundaries where it maintains transcriptional activity. This binding is essential for silencing satellite repeats, preventing DNA damage and genome instability. PHF2 depletion increases the transcription of heterochromatic repeats, accompanied by a decrease in H3K9me3 levels and alterations in PcH organization. We further show that PHF2's PHD and catalytic domains are crucial for maintaining PcH stability, thereby safeguarding genome integrity. These results highlight the multifaceted nature of PHF2's functions in maintaining heterochromatin stability and regulating gene expression during neural development. Our study unravels the intricate relationship between heterochromatin stability and progenitor proliferation during mammalian neurogenesis.

EMBO press papers are accompanied online by A) a short (1-2 sentences) summary of the findings and their significance, B) 2-3 bullet points highlighting key results and C) a synopsis image that is exactly 550 pixels wide and 200-600 pixels high (the height is variable). The synopsis image should provide a sketch of the major findings, like a graphical abstract. Please note that text needs to be readable at the final size. Please send us this information along with the final manuscript.

Referee #1:

Within the revised version, the authors have significantly improved the manuscript on a range of issues. As such, I feel the data better support the claims that were made.

However, there are still several issues left with respect to data/analysis or interpretations that in my opinion require serious attention:

- During the revision, the authors added two additional PHF2 ChIP-seq replicates, which is helpful. However, in many aspects the four replicates that are presented now show critical differences. For example, in Fig 1c the enrichment over "LTRs" and "DNA", and to a lesser extent over "SINEs" and "LINEs", is different between the replicates. Also, a simple view on the data (Figure S1E and EV1c) shows considerable differences between the replicates: In Fig S1e replicate 3 is strongly deviating, while in Fig EV1c, there is inconsistency between all replicates. This is supported by the correlation plot in Fig S1d showing a poor correlation between R1/R2 versus R3/R4. As such, it remains unclear which of these profiles represent bona fide PHF2 binding. The authors should consider removing low quality profiles, and properly explain in the manuscript how the observed discrepancies affect their further analysis throughout the paper.
- Also, in the PHF2 ChIP-qPCR in Fig S1c (top panel) it is indicated that Col2a1 shows non-significant differences (even referred to in the text). However, based on eyeballing, there is a very clear difference between PHF2 and IgG (the latter does not show any signal). Also, the bottom panel does show signals in IgG. Again, big differences in PHF2 between replicates (not even mentioning the %input levels being very different), here in ChIP-qPCR. Together with the ChIP-seq, this makes me concerned about data reproducibility and quality.
- The interaction mass spectrometry data is still not properly explained, analyzed, and represented. The table with accession numbers and scores is not helpful for reading and re-use. As is common in the field, the authors might want to represent the data in a volcano plot, in which the PHF2 IP is plotted over negative IP, with p-values and fold changes indicated on the axis. Also, protein names are much more human interpretable as compared to accession numbers. Additionally, a proper table with LFQ or iBAQ values of all experiments is standard and helps for better interpretation.
- In Fig EV1a, the authors indicate and write in the text that H3K9me3 is enhanced over the various repeat elements. However, the median levels of H3K9me3 for all repeats seem to be around 0, suggesting to me just background levels (no depletion, but also no enrichment). This is confusing.
- For the RNA-seq analysis in boundary levels, I find it counterintuitive to plot average profiles (Fig 5e). While for ChIP-seq this might be informative, the nature by which RNA is expressed (from very discrete places (genes) in the genome) can result in misleading interpretations when averaging out signals. The authors should look at discrete genes or discrete expressed loci, and compare these.
- This paper, as it represents a somewhat more difficult concept, would highly benefit from ending with a graphical representation (model), which is now removed during revisions.

Referee #2:

I wish to express my sincere appreciation for the dedication and hard work exhibited by all the co-authors. They substantially enhanced both the manuscript and the figures. Their meticulous consideration of the feedback and suggestions provided by the three referees is greatly valued and acknowledged.

I have only one question. In Appendix figure S1 E (IGV plot with all the 4 replicates). The replicate n{degree sign}3 seems quite different from the others (more peaks?). So I wonder how it was managed in the analysis.

So I recommend this manuscript for publication in EMBO reports.

Referee #3:

The authors have carefully revised their manuscript, including new qPCR and imaging data, which makes the manuscript stronger. They have also adjusted their model/discussion that now fits better to their data. In general, they have addressed my main concerns, and I recommend publication.

As an additional remark to the authors, beyond the scope of the current manuscript: To further stratify the potential models and to see if heterochromatin components are titrated out by novel genomic binding sites, the authors might want to consider overexpressing these heterochromatin components (for example HP1) to test if this can rescue/attenuate the observed phenotypes.

Cross-comments from referee 2:

I agree with referee 1 regarding the differences between Replicates R1/R2 and R3/R4. That's why I was inquiring about how the authors managed R3 (which appears to be deviant) in the analysis. Thus, there are still some improvements to be made, and one potential solution could be to remove R3 and re-do the statistical analysis. The most effective way to address this issue would be to add a fifth replicate (in place of the R3), but that would require a new experiment (which would require additional time), so I'm not sure you would allow it.

I am not qualified to judge the Mass Spec data, so I will defer to the advice of referee 1.

Similar to referees 1 and 3, I agree that a graphical model would enhance the understanding of the biological hypothesis, particularly as the role of PHF2 described here appears counterintuitive. It will be easier to implement because it was in the previous version of the manuscript, although it will need some modification to fit completely with the hypothesis.

Cross-comments from referee 3:

> However, there are still several issues left with respect to data/analysis or interpretations that in my opinion require serious attention:

> - During the revision, the authors added two additional PHF2 ChIP-seq replicates, which is helpful. However, in many aspects the four replicates that are presented now show critical differences. For example, in Fig 1c the enrichment over "LTRs" and "DNA", and to a lesser extent over "SINES" and "LINEs", is different between the replicates. Also, a simple view on the data (Figure S1E and EV1c) shows considerable differences between the replicates: In Fig S1e replicate 3 is strongly deviating, while in Fig EV1c, there is inconsistency between all replicates. This is supported by the correlation plot in Fig S1d showing a poor correlation between R1/R2 versus R3/R4. As such, it remains unclear which of these profiles represent bona fide PHF2 binding. The authors should consider removing low quality profiles, and properly explain in the manuscript how the observed discrepancies affect their further analysis throughout the paper.

Ref 3: I agree that there is variation between the replicates, and it might be worth asking the authors about it. Regarding the main figures, the authors write "Our results indicate that PHF2 binding sites were enriched at specific repetitive elements such as satellites and long interspersed nuclear elements (LINEs) (Fig. 1C)", which seems to be justified by the data in Fig. 1C (all 4 replicates show an enrichment of satellites and LINEs -> white-to-red color range).

> - Also, in the PHF2 ChIP-qPCR in Fig S1c (top panel) it is indicated that Col2a1 shows non-significant differences (even referred to in the text). However, based on eyeballing, there is a very clear difference between PHF2 and IgG (the latter does not show any signal). Also, the bottom panel does show signals in IgG. Again, big differences in PHF2 between replicates (not even mentioning the %input levels being very different), here in ChIP-qPCR. Together with the ChIP-seq, this makes me concerned about data reproducibility and quality.

Ref 3: I also agree with this observation by referee #1. However, I personally wonder if this might be a "plotting error" in Fig. S1c, because I do not see how the displayed difference between PHF2 and IgG can be "not significant" in the upper plot, based on the error bar of the PHF2 ChIP (green) and the absence of any visible signal and error in the IgG ChIP. It rather looks like the bar for the IgG ChIP is simply not plotted. The authors should indeed correct this.

> - The interaction mass spectrometry data is still not properly explained, analyzed, and represented. The table with accession numbers and scores is not helpful for reading and re-use. As is common in the field, the authors might want to represent the data in a volcano plot, in which the PHF2 IP is plotted over negative IP, with p-values and fold changes indicated on the axis. Also, protein names are much more human interpretable as compared to accession numbers. Additionally, a proper table with LFQ or iBAQ values of all experiments is standard and helps for better interpretation.

Ref 3: I agree that providing iBAQs would be helpful if the authors can do this. However, the MS analysis is not really a focus of the paper, and the authors mainly use it to motivate their experiment with HP1BP3. From my perspective, it is okay as is, although the raw data should be deposited, which the authors claimed to have done (I have not checked it myself).

> - In Fig EV1a, the authors indicate and write in the text that H3K9me3 is enhanced over the various repeat elements.

However, the median levels of H3K9me3 for all repeats seem to be around 0, suggesting to me just background levels (no depletion, but also no enrichment). This is confusing.

Ref 3: I agree with the referee, this is written in a confusing way. I guess the authors should write that the repeats bound by PHF2 according to Fig. 1C (satellites, LINEs) carry more H3K9me3 than other repeats that reside in more open euchromatin (for example SINEs). For such statements, and in the absence of spike-ins, it all depends on what you compare to what. The genomic average will not be "background" as a significant part of the genome probably contains H3K9me3 (at least this is the case in fibroblasts), so "0" after input subtraction probably means that there is H3K9me3 on these repeats, with the level corresponding to the average level found across the genome (assuming that RPKM values are used for ChIP and input).

> - For the RNA-seq analysis in boundary levels, I find it counterintuitive to plot average profiles (Fig 5e). While for ChIP-seq this might be informative, the nature by which RNA is expressed (from very discrete places (genes) in the genome) can result in misleading interpretations when averaging out signals. The authors should look at discrete genes or discrete expressed loci, and compare these.

Ref 3: I think it can be instructive to look at RNA-seq data in larger bins, especially if the same bin (the same group of genes) is compared across conditions (shCT versus shPHF2). I agree that each value will be a weighted average over the genes in this bin, so information at the level of genes will be averaged out/lost.

> - This paper, as it represents a somewhat more difficult concept, would highly benefit from ending with a graphical representation (model), which is now removed during revisions.

Ref 3: In principle I agree, but this should be left to the authors. As I found that the model in the first version did not reflect the data very well, I personally prefer to have no model than the original one.

ANSWER TO THE EDITOR AND THE REVIEWERS

EMBOR-2023-58575V2 "PHF2-mediated H3K9me balance orchestrates heterochromatin stability and progenitor proliferation in early neurogenesis"

We sincerely appreciate both the editor's and the reviewers' comments. We have now provided an answer to the referee's comments following the suggestions made by the editorial.

We include below a detailed, "point-by-point" response to them. Reviewer's suggestions are printed in Times bold, and our replies are printed in Times.

For the sake of simplicity, we decided to address together the comments from all the referees in the "Cross comments from referee" section.

REFEREE 1

We thank Reviewer 1 for their constructive comments. We acknowledge the significance of the raised issues and, in response to their recommendations as well as the editorial's, we have made changes to present our results more clearly and consistently.

1. During the revision, the authors added two additional PHF2 ChIP-seq replicates, which is helpful. However, in many aspects the four replicates that are presented now show critical differences. For example, in Fig 1c the enrichment over "LTRs" and "DNA", and to a lesser extent over "SINES" and "LINEs", is different between the replicates. Also, a simple view on the data (Figure S1E and EV1c) shows considerable differences between the replicates: In Fig S1e replicate 3 is strongly deviating, while in Fig EV1c, there is inconsistency between all replicates. The authors should consider removing low quality profiles, and properly explain in the manuscript how the observed discrepancies affect their further analysis throughout the paper.

Cross-comments from referee 2:

I agree with referee 1 regarding the differences between Replicates R1/R2 and R3/R4. That's why I was inquiring about how the authors managed R3 (which appears to be deviant) in the analysis. Thus, there are still some improvements to be made, and one potential solution could be to remove R3 and re-do the statistical analysis.

Cross-comments from referee 3:

I agree that there is variation between the replicates, and it might be worth asking the authors about it. Regarding the main figures, the authors write "Our results indicate that PHF2 binding sites were enriched at

specific repetitive elements such as satellites and long interspersed nuclear elements (LINEs) (Fig. 1C)", which seems to be justified by the data in Fig. 1C (all 4 replicates show an enrichment of satellites and LINEs -> white-to-red color range).

Response:

We really appreciate the feedback provided by Referee 1 on the revised version of our manuscript. We thank the referees for their comments about our replicates, and we agree that they are different to some extent.

As explained in the previous revision, replicates 3 and 4 were performed later, as requested during the manuscript review, using a different batch of antibody. Thus, R1/R2 and R3/R4 samples have been generated in different years and by employing distinct antibodies, which can contribute to increase the variability between replicates. This is why the Pearson correlation, while acceptable, is lower than in the previous version where we used two replicates performed with the same antibody and in parallel.

Regarding the correlation plot (Fig. S1D), we do not think that a correlation coefficient between 0.58 and 0.72 represents "a poor correlation", since 0.6-0.8 values are normally considered to indicate a moderate/strong correlation. In addition, although we noticed a slightly variability in our replicates, we repeated all the analysis with the novel samples both alone and also merging them with the old ones, obtaining reproducible results with regard to the PHF2 enrichment within Satellite repeats and pericentromeric regions, which are the two main points of our study. These are the main reasons why we decided to proceed with the analysis including the new R3 and R4 replicates: we knew that the differences were probably due to added noise by the use of a different antibody in a different year, but it did not affect to the results and main conclusions of the presented investigation. Nevertheless, we coincide that it would be beneficial to select the most suitable samples for the analysis, therefore reducing the noise of the calculations. As suggested by Referees 1 and 2, we decided to discard replicate with lower quality (Pearson correlation coefficient 0.58) and redo the corresponding analyses, which are now included in the current version of the paper.

During the reevaluation of our replicates, we detected human errors in the assignment of the screenshots to their corresponding replicate. These errors have been corrected in the revised version of the manuscript.

2. Also, in the PHF2 ChIP-qPCR in Fig S1c (top panel) it is indicated that Col2a1 shows non-significant differences (even referred to in the text). However, based on eyeballing, there is a very clear difference between PHF2 and IgG (the latter does not show any signal). Also, the bottom panel does show signals in IgG. Again, big differences in PHF2 between replicates (not even mentioning the %input levels being very different), here in ChIP-qPCR.

Cross-comments from referee 3:

I also agree with this observation by referee #1. However, I personally wonder if this might be a "plotting error" in Fig. S1c, because I do not see how the displayed difference between PHF2 and IgG can be "not significant" in the upper plot, based on the error bar of the PHF2 ChIP (green) and the absence of any visible signal and error in the IgG ChIP. It rather looks like the bar for the IgG ChIP is simply not plotted. The authors should indeed correct this.

Response:

We completely agree with Referee 1 and apologize for the error. The IgG ChIP bar was not plotted, but this error has now been corrected.

The reason for the different enrichments is again due to replicates being performed with a different antibody. However, the initial results are confirmed by all four replicates.

3. The interaction mass spectrometry data is still not properly explained, analyzed, and represented. The table with accession numbers and scores is not helpful for reading and re-use. As is common in the field, the authors might want to represent the data in a volcano plot, in which the PHF2 IP is plotted over negative IP, with p-values and fold changes indicated on the axis. Also, protein names are much more human interpretable as compared to accession numbers. Additionally, a proper table with LFQ or iBAQ values of all experiments is standard and helps for better interpretation.

Cross-comments from referee 3:

I agree that providing iBAQs would be helpful if the authors can do this. However, the MS analysis is not really a focus of the paper, and the authors mainly use it to motivate their experiment with HP1BP3. From my perspective, it is okay as is, although the raw data should be deposited, which the authors claimed to have done (I have not checked it myself).

Response:

We value the referee's suggestion and completely agree that a volcano plot and statistical data would enhance our analysis. However, as discussed with the editor and referees at the beginning of the review process, the data suggesting that PHF2 might interact with HP1BP3 emerged from a single replicate of mass spectrometry analysis. Due to that, fold change values and their respective FDR or p-values cannot be computed, which consists of the

required data to construct a Volcano plot.

In addition, and as accurately highlighted by Referee 3, we only used the MS data to introduce the experiment with HP1BP3. The manuscript's objective was not to showcase the entire PHF2 proteome but to demonstrate PHF2's interaction with a specific heterochromatin protein, HP1BP3, as confirmed by CoIP. Our intention is solely to illustrate the interaction between these two factors, not to conduct a comprehensive study of the entire PHF2 proteome. After consulting with the editor, we have deposited the raw data and presented the entire

interactome in Appendix Table 1. The accession number is included in the revised version of the manuscript in the Data Availability section.

Following Referee 1's suggestion and to facilitate the analysis of the mass spectrometry results, we have replaced the accession numbers with protein names in Table S1

4.- In Fig EV1a, the authors indicate and write in the text that H3K9me3 is enhanced over the various repeat elements. However, the median levels of H3K9me3 for all repeats seem to be around 0, suggesting to me just background levels (no depletion, but also no enrichment). This is confusing.

Cross-comments from referee 3:

I agree with the referee, this is written in a confusing way. I guess the authors should write that the repeats bound by PHF2 according to Fig. 1C (satellites, LINEs) carry more H3K9me3 than other repeats that reside in more open euchromatin (for example SINEs). The genomic average will not be "background" as a significant part of the genome probably contains H3K9me3 (at least this is the case in fibroblasts), so "0" after input subtraction probably means that there is H3K9me3 on these repeats, with the level corresponding to the average level found across the genome (assuming that RPKM values are used for ChIP and input).

Response:

We appreciate the feedback from the referees and have revised the sentence for clarity. As suggested by the Referee 3, the modified text now indicates: "As expected, the sequences bound by PHF2 according to Fig.1C were depleted for H3K4me3 and presented higher levels of H3K9me3 than other repeats that reside in more euchromatic regions, such as SINEs (Fig. EV1A)".

Regarding the interpretation of average quantification values close to "0", we also consider that this does not necessarily indicate background levels, as clarified by the Referee 3. Additionally, it's important to note that the classes of repeats used for this analysis encompass thousands to millions of repeats (samples sizes: Satellite = 36,234; LTR = 970,039; LINE = 987,285; SINE = 1,527,608; DNA = 162,787). We did not observe significant enrichment of H3K9me3 across all the repeats included in each class, as they are probably regulated by different epigenetic mechanisms. However, upon analyzing different families within the Satellite class, we did observe a clear enrichment in some groups (e.g., GSAT_MM), which specifically correspond to the regions of interest in this study: mouse major satellite repeats.

5.- For the RNA-seq analysis in boundary levels, I find it counterintuitive to plot average profiles (Fig 5e). While for ChIP-seq this might be informative, the nature by which RNA is expressed (from very discrete places (genes) in the genome) can result in misleading interpretations when averaging out signals. The authors should look at discrete genes or discrete expressed loci, and compare these.

Cross-comments from referee 3:

I think it can be instructive to look at RNA-seq data in larger bins, especially if the same bin (the same group of genes) is compared across conditions (shCT versus shPHF2). I agree that each value will be a weighted average over the genes in this bin, so information at the level of genes will be averaged out/lost.

Response:

We thank the referees for their feedback on our RNA-Seq analysis. As highlighted by Referee 3, we think that this type of analyses can be informative, since our primary aim was to explore the correlation between global transcription and epigenetic changes in shCT versus shPHF2 conditions. In fact, this analysis revealed a decrease in RNA-Seq quantification concurrent with a widespread increase in H3K9me3 across the pericentromeric regions of the chromosomes. Furthermore, it has been previously documented that transcription plays a role in the establishment and regulation of pericentromeric regions. However, these regions are not notably enriched in coding genes, and RNA-Seq data can capture the expression of some non-coding regions such as repeats, lncRNAs, and tRNAs. For these reasons, we decided to explore global gene expression measured by RNA-Seq using larger bins.

6.- This paper, as it represents a somewhat more difficult concept, would highly benefit from ending with a graphical representation (model), which is now removed during revisions.

Cross-comments from referee 2:

Similar to referees 1 and 3, I agree that a graphical model would enhance the understanding of the biological hypothesis, particularly as the role of PHF2 described here appears counterintuitive. It will be easier to implement because it was in the previous version of the manuscript, although it will need some modification to fit completely with the hypothesis

Cross-comments from referee 3:

In principle I agree, but this should be left to the authors. As I found that the model in the first version did not reflect the data very well, I personally prefer to have no model than the original one.

Response:

The graph presented in previous versions was removed at the suggestion of Referee 3 because, as indicated, it was not accurate.

Following the referee's and editorial's suggestions, we have created a model that recapitulates the new results and interpretation, which will be included as a graphical abstract

REFEREE 2

I wish to express my sincere appreciation for the dedication and hard work exhibited by all the co-authors. They substantially enhanced both the manuscript and the figures. Their meticulous consideration of the feedback and suggestions provided by the three referees is greatly valued and acknowledged.

I have only one question. In Appendix figure S1 E (IGV plot with all the 4 replicates). The replicate n{degree sign}3 seems quite different from the others (more peaks?). So I wonder how it was managed in the analysis.

Response:

We express our gratitude to Reviewer 2 for providing constructive and positive feedback, which I am confident has helped improve the quality of our work.

Following your recommendation, as well as that of Reviewer 1, we have removed the PHF2 ChIP-seq with lower Pearson correlation coefficient (0.58) and redo the corresponding analyses.

REFEREE 3

The authors have carefully revised their manuscript, including new qPCR and imaging data, which makes the manuscript stronger. They have also adjusted their model/discussion that now fits better to their data. In general, they have addressed my main concerns, and I recommend publication.

Response:

I would like to express my gratitude to Referee 3 for their constructive comments, which have prompted us to reflect on certain aspects of our work, particularly the interpretation of the results. I am confident that these comments have contributed to the overall improvement of our work, and we wish to extend our thanks.

Dr. Marian Martínez-Balbas
CSIC
Structural and Molecular Biology
Baldiri i Reixac 15-21
Barcelona, Barcelona 08028
Spain

Dear Marian,

I am very pleased to accept your manuscript for publication in the next available issue of EMBO reports. Thank you for your contribution to our journal.

I would like to suggest to slightly modify the title to:

PHF2-mediated H3K9me balance orchestrates heterochromatin stability and neural progenitor proliferation

"Neural development" in the title is slightly misleading, I think.

Also, please make sure that all deposited data will be freely accessible upon online publication of your manuscript (in approximately 2 weeks).

If you have any questions, please do not hesitate to contact the Editorial Office. Thank you very much for your contribution to EMBO Reports.

Best wishes,
Esther

Referee #1:

In line with the other referees, I like to compliment the authors by their clarity, dedication and hard work, also during revision. The authors have carefully considered and addressed my remaining issues, further improving the manuscript. As such, I recommend publication in EMBO Reports.
